# A microphysiological human mini-bladder reveals urine-urothelium interplay in tissue resilience and UPEC recurrence in urinary tract infections

Gauri Paduthol [1,8], Mikhail Nikolaev [2,8], Kunal Sharma [1], Jérôme Blanc[3], Kathrin Tomasek [1], Léa Ivana Esméralda Schlunke [1], Valentin Borgeat [1], Giovanna Ambrosini [4], Irina Kolotuev[5], Stéphanie Clerc-Rosset[3], Nikolche Gjorevski [2], Graham W. Knott [3], Matthias P. Lütolf[2,6,9] ✉, Vivek V. Thacker [7,9] ✉ & John D. McKinney [1,9] ✉

Urine is a dynamic and highly variable biofluid. Urine-urothelium interactions are a critical yet underexplored factor in bladder homoeostasis and urinary tract infections (UTIs). Here, we report on a human 'mini-bladder' model that exposes a stratified urothelium to urine of defined composition, and incorporates micturition. Prolonged exposure to high-solute concentration urine weakens tight junctions, dysregulates immune responses, and reduces bladder tissue resilience. This increases susceptibility to colonisation of the bladder by uropathogenic *Escherichia coli* (UPEC) which reduces efficacy of antibiotic therapy. In high-solute concentration urine, Fosfomycin monotherapy – prescribed for uncomplicated UTIs, induces the formation of cell wall-deficient (CWD) UPEC in the urine (as observed in patients with recurrent UTIs) but also within deeper urothelial layers. Tissue-associated CWD UPEC directly contributes to recurrence. Our findings expand the conceptual role for CWD UPEC in UTIs, and demonstrate the power of the mini-bladder platform to capture urine-urothelial microenvironment dynamics that actively shape UTI pathogenesis and antibiotic tolerance.

The bladder is a highly dynamic organ that undergoes substantial mechano-physiological fluctuations[1–3] while retaining urine of varying composition and toxicity[4,5]. These fluctuations are accommodated by the bladder epithelium that functions as a critical mucosal barrier. The bladder epithelium actively modulates urine composition[6–8], absorbing more salts and urea in dehydrated conditions[9–12]. In experiments in animal models, including rodents, urine with high concentrations of solutes in turn damages the integrity of the urothelium, causing tight

[1]Laboratory of Microbiology and Microtechnology, School of Life Sciences (SV), Ecole Polytechnique Fédérale de Lausanne, Lausanne, Switzerland. [2]Institute of Human Biology (IHB), Roche Pharma Research and Early Development, Roche Innovation Center Basel, Basel, Switzerland. [3]BioElectron Microscopy Facility, Ecole Polytechnique Fédérale de Lausanne, Lausanne, Switzerland. [4]Bioinformatics Competence Centre, École Polytechnique Fédérale de Lausanne, Lausanne, Switzerland. [5]Department of Biomedical Sciences, Université De Lausanne, Lausanne, Switzerland. [6]Institute of Bioengineering, School of Life Sciences (SV), Ecole Polytechnique Fédérale de Lausanne, Lausanne, Switzerland. [7]Department of Infectious Diseases, Medical Microbiology and Hygiene, Medical Faculty Heidelberg, Heidelberg University, Heidelberg, Germany. [8]These authors contributed equally: Gauri Paduthol, Mikhail Nikolaev. [9]These authors jointly supervised this work: Matthias P. Lütolf, Vivek V. Thacker, John D. McKinney. ✉e-mail: matthias.lutolf@roche.com; vivek.thacker@uni-heidelberg.de; john.mckinney@epfl.ch

junction blebbing, bladder interstitial congestion, increased apoptosis signalling and general irritation[13–16]. An effective and long-term colonisation of the bladder by pathogens depends on a dynamic interaction between the urothelium and urine. It is increasingly evident that this urothelial niche promotes persistence and antibiotic tolerance in urinary tract infections (UTIs)[17–20], which characteristically have a high recurrence rate in susceptible individuals[21].

The impact of urothelial niches and bladder dynamics is evident in the life cycle of uropathogenic *Escherichia coli* (UPEC), the most common causative pathogen of UTIs[22]. UPEC has developed multifaceted survival and propagation strategies within the complex and variable environment of urine and the urothelium. UPEC proliferates in urine, colonises the urine microbiome, and adheres strongly to the urothelium[23]. A subset of this bacterial population enters superficial umbrella cells, where they can grow to form large reservoirs as intracellular bacterial communities (IBCs) that are shielded from the immune system[24–26]. Consequently, the upper cell layers of the urothelium actively exfoliate to reduce the bacterial load on the tissue, releasing part of the invaded UPEC back into the urine[27]. Deeper in the tissue, UPEC can survive in smaller intracellular clusters termed quiescent intracellular reservoir (QIRs) long after superficial inflammation has ceased[18,28,29]. More recent work has shown that the bladder epithelium can be rapidly populated by solitary bacteria that are intracellular or even pericellular[30]. Finally, UPEC in urine samples from patients treated with cell wall targeting antibiotics such as Fosfomycin have been isolated in cell wall deficient (CWD) L-forms[31–33]. These are particularly prominent in the urine of recurrent UTI (rUTI) patients[32], suggesting a link between CWD forms and re-infection of tissue when external stressors are released. This large range of bacterial forms within a single tissue highlights the need to understand pathogenesis and therapy in a dynamic fashion across two very different microenvironments in direct interaction with each other—first, a stratified urothelium and second, urine with variable composition and solute concentration. Examination of bacteria in the tissue is only possible as an endpoint measurement in animal models, which limits dynamical information, whereas human urine has a wide range of solute concentrations (osmolarity) across healthy[34] and diseased[35] individuals, which is difficult to replicate in mice under laboratory conditions.

In recent years, in-vitro microphysiological models that replicate aspects of the bladder microenvironment have been used to study UTIs[36–39]. For example, the bladder organoid model, albeit a closed system, offers a stratified tissue with medium-throughput and possibilities to incorporate immune cells[30]. This model is complemented by the bladder-on-chip that lacks stratification but incorporates micturition cycles, exposure to diluted pooled urine and vascular flow[40]. Building on this, the 3D urine-tolerant human urothelial (3D-UHU) model has a robust stratified tissue that remains viable when exposed to undiluted pooled urine, although it doesn't recapitulate micturition effects[41]. These models have complementary strengths and weaknesses, but so far, no single model incorporates a stratified, well-differentiated urothelium, an accessible lumen for urine perfusion and micturition, and the ability to stretch the tissue[42]—the three features that in combination are needed to capture the interaction of urine, urothelium and the respective niches formed by UPEC.

Here, we present a human 'mini-bladder' organoid model that meets these requirements. The tissue has multi-layer stratification and polarised differentiation, with an apical layer of umbrella cells and a tight tissue barrier exposed to urine. Micturition effects are established through induced urine flow or pressure-controlled tissue stretching. Consistent with data from animal models[14,16,43], long-term exposure of the mini-bladder urothelium to urine of high solute concentration supresses epithelial immune responses, reduces barrier function and increases the incidence of cell death. Through high-resolution imaging of untreated infections over several micturition cycles, we capture the dynamics of interactions between UPEC in urine

(whose solute concentration can be controlled) and the bladder tissue. Interestingly, we show that long-term exposure to urine with high solute concentration reduces tissue resilience to infection, leads to an increased accumulation of tissue-associated UPEC and reduced efficacy of antibiotic clearance. Next, we explore the differential behaviours of these bacterial sub-populations to antibiotic therapy. Cell wall targeting antibiotics like Fosfomycin generate cell-wall deficient (CWD) UPEC survivors in high concentration urine in the mini-bladder, replicating the observation from rUTI patients' urine in a microtissue environment. We demonstrate a direct role for CWD UPEC in the urine for reseeding infections, and unexpectedly, we find that CWD UPEC that survive antibiotic can also be found in the deeper layers of the urothelium, revealing a new mechanism in the tissue that may contribute to the recurrence of infection. Collectively, this study shows that the mini-bladder serves as a powerful in vitro model for capturing the complete interplay between tissue and urine in UTIs.

## Results

### Mini-bladder recreates human urothelial architecture with tissue stretch and urine flow

We developed a human bladder microtissue model built on the hydrogel-based perfusable microfluidic architecture derived from Nikolaev et al.[44] (Fig. 1A, Supplementary Fig 1A–D). Primary human bladder epithelial cells were seeded into the central channel of the device, and once a stratified tissue was obtained at day 15, this was exposed to urine on the apical side and nutritive media on the basal side for a further 5–7 days (Supplementary Fig 1E). The resultant tissue at day 20 (Fig. 1B, Supplementary Movie 1, Supplementary Movie 2) recreates the organisation of the human bladder epithelium. Accurate recreation of umbrella cells is critical to ensure stability of bladder tissue when exposed to urine[45]. In the mini-bladder, cells on the apical layer proximal to the lumen were CK20+ (Fig. 1 C, Supplementary Fig 2A, C), UP3A+ (Supplementary Fig 2D), UP1A+ (Supplementary Fig 2E), and in several instances bi-nucleated with honeycomb like structure (Supplementary Fig 2B, Fig. 1D), consistent with the known characterisation of the superficial umbrella cell layer[45]. In contrast, cells nearer the hydrogel scaffold expressed the basal/intermediate cell marker CK8+[46] (Fig. 1E, Supplementary Fig 2F). CK13+ cells[47], marking the differentiation of basal-like squamous to a transitional differentiated cells in vivo, were also observed within the stratified tissue (Fig. 1E, Supplementary Fig 2A, G). The increased expression of *UP3A*[48], *KRT20*, *KRT8* and *KRT13* in the mini-bladder relative to the cells seeded into the device was confirmed by qRT-PCR analyses as well (Fig. 1F). An upregulation of *KRT14*[49], a marker for a subset of basal cells capable of self-renewal and *TP63*[50], a common progenitor cell marker, was also observed (Fig. 1 F). The mini-bladder model, therefore, has appropriate stem-cell niches required for urothelial renewal, relevant for modelling the tissue response to infection or sterile injury. The stratified tissue maintains a robust barrier; strong ZO1[51] staining is observed on the entire apical surface lining the lumen (Fig. 1G, Supplementary Fig 2H). Barrier integrity was also confirmed by a dextran dye diffusion test (Fig. 1H). The diffusion of dextran out of the lumen into the hydrogel was 92.8 % lower in a fully differentiated mini-bladder compared to an empty device, and the diffusion decreased over time, consistent with the differentiation of a stratified tissue from a monolayer (Fig. 1I).

The robust urothelium in the mini-bladder model could be subjected to changes in bladder volume mimicking micturition by controlled hydraulic pressure differences between the inlet and outlet (S. Mov-3). When stretched over 30 s, the urothelial physiology was rapidly altered from larger cuboidal cells to a tightly-packed structure with thinner, flattened apical cells (Fig. 1J, Supplementary Fig 3A, B), in good agreement with images from stretched and relaxed bladder explants[52–54]. Furthermore, tight junction contacts are evident in the lumenal cells (Supplementary Fig 3C), and this is enhanced overall in

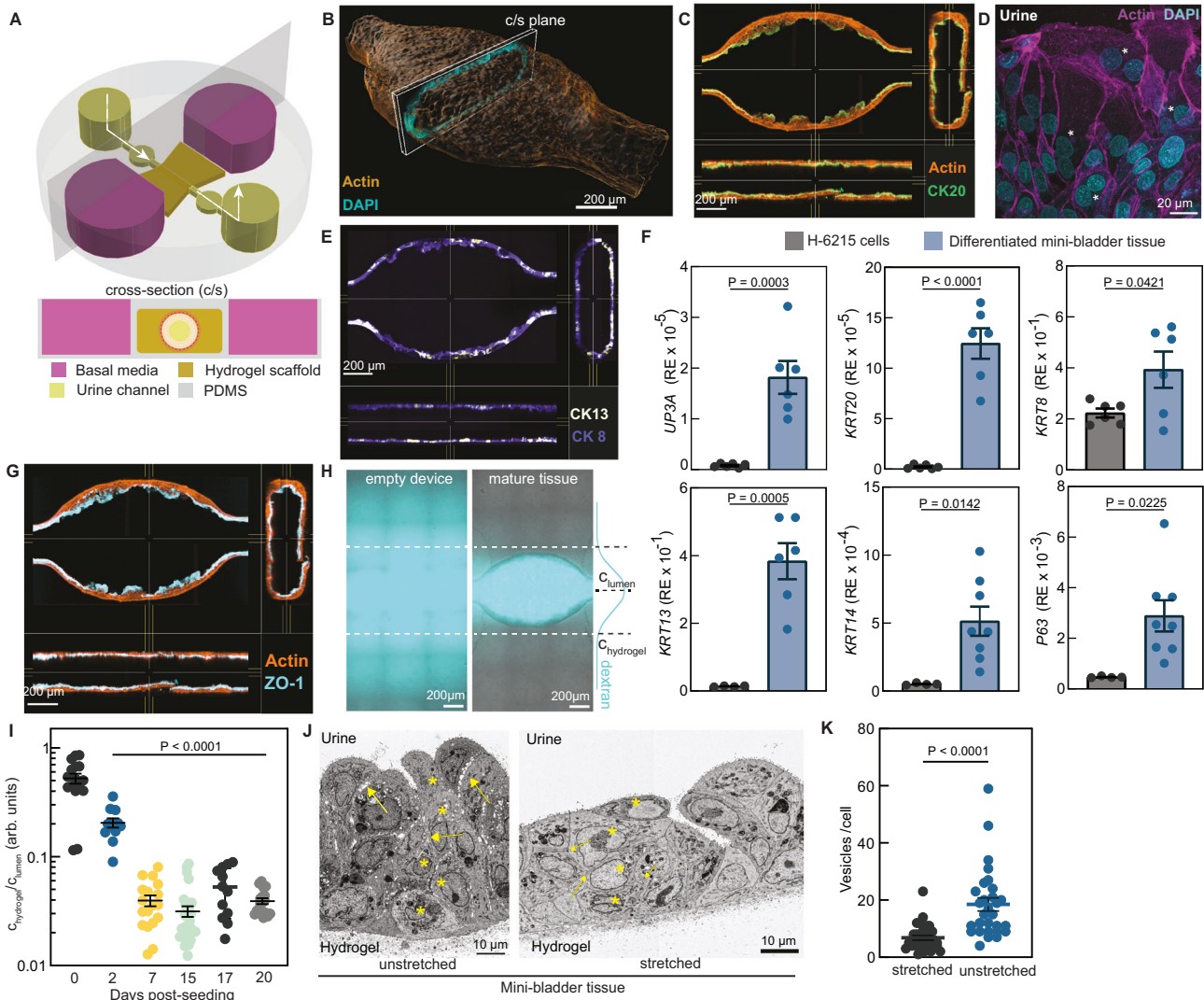

**Fig. 1 | Human mini-bladder model of rUTI recapitulates the mechano-physiology of the urothelium. A** Schematic of the human mini-bladder model. The PDMS device (grey) contains a hydrogel chamber (brown) flanked by chambers filled with basal media (pink). The cross-sectional view shows how the hydrogel chamber is etched in the centre to connect the inlet and the outlet (yellow). The (white) arrows indicate the direction of lumenal flow. **B** 3D reconstruction of the microtissue (actin, amber), the central cross section shows nuclei arrangement (cyan). **C** Representative orthogonal view of microtissue shows CK20 (green) and actin (amber). **D** Representative microtissue cross-section shows binucleated cells in white asterisks (actin, magenta; DAPI, cyan). **E** Representative orthogonal view of microtissue shows CK13 (purple) and CK8 (yellow). **F** Relative expression (RE) of indicated genes by qRT-PCR in the epithelial cells seeded in the mini bladder on day 0 ($n = 3$ biological replicates) and the mature mini bladder tissue on day 20 ($n = 3$

biological replicates). **G** Representative orthogonal view of microtissue shows ZO-1 (cyan) and actin (amber). **H** Representative overlay of bright-field and fluorescence images for dextran diffusion assay in an empty mini-bladder device and a differentiated mini-bladder microtissue at day 20. (Fluorescein isothiocyanate (FITC)-tagged dextran, cyan). **I** Quantification of relative dextran permeability from (**H**), $n = 3, 2, 3, 5, 2$, and $3$ biological replicates at day 0, 2, 7, 15, 17 and 20, respectively. **J** Array tomography SEM images of unstretched and stretched tissue cross-sections. Cell to cell junctions marked by yellow arrows; tissue stratification indicated by yellow asterisks. **K** Quantification of the number of vesicles per cell from (**J**) (number of cells, $n = 36$ for stretched, $n = 29$ for unstretched). Data represented as mean ± SEM. *P*-values calculated using a two-sided unpaired t-test in (**F**, **I**, **K**). Images in (**C**–**E**, **G**) are representative of at least 3 biologically independent experiments. Source data are provided as a Source data file.

the tissue upon stretch[53] (Fig. 1J). One mechanism by which the bladder adapts to rapid volume changes is through the presence of cytoplasmic vesicles in the umbrella cells that have been shown to transport to and fuse with the apical membrane to increase its surface area upon bladder filling[52]. Consistently, we observed the presence of several cytoplasmic vesicles in the unstretched tissue (Supplementary Fig 3C, D) and an overall reduction in the number of vesicles in the cytoplasm post-stretching (Fig. 1K).

Together, these results confirm that the mini-bladder microtissue model contains a stratified, differentiated urothelium exposed to urine whose morphology can react to mechanical stretching in a manner consistent with in vivo characterisations of bladder tissue.

## Urine with high solute concentration alters tissue integrity and epithelial immune response

The urothelium in the mini-bladder can be stably maintained for several days in co-culture with urine. This allowed us to study how exposure to urine of variable composition over several days impacts the state of the bladder epithelium. Although not typically considered a factor in pathogenesis, human urine has a range of osmolarities, pH and compositions that vary widely between individuals over time and is very sensitive to overall health[4,5,55]. Over a period of 9 days, mini-bladders seeded with the same initial cells and differentiated under the same conditions showed different responses when exposed to urine pooled from different individuals (Supplementary Fig 4A). Exposure to

pooled human urine (PHU) with high solute concentrations (Supplementary Table 1) typically led to accelerated tissue damage, as evidenced by an increase in the proportion of DRAQ7+ dead cells and a reduction in the overall cell numbers (Supplementary Fig 4B, C).

Ipe et al.[56] developed a protocol for synthetic human urine (SHU) that closely mimics human urine in nutritional content but with well-defined ingredients. This formulation has been shown to support the growth of several uropathogens and has been used as a surrogate to pooled human urine to model antibiotic responses in UTI[56,57]. We hypothesised that the major driver of the variability in tissue responses to different PHU samples was the differing solute concentration, quantified through osmolarity. Accordingly, we developed two synthetic human urine formulations with osmolarities adjusted to match the upper and lower range in the pooled human urine samples (Supplementary Table 1). Both formulations contain the same amount of casamino acids, which reflects in the similar doubling time of UPEC in both formulations and are at an acidic pH = 5.6 for long-term stability. Prior to use in the mini-bladder model, the pH is adjusted (with 20 mM HEPES, 1 mM CaCl$_2$) to above 6 to match the range of PHU. The two formulations are termed high solute and low solute SHU, respectively and are used for experiments in the rest of the manuscript unless otherwise described[58]. Consistent with results in PHU, mini-bladders exposed to high solute SHU showed a gradual increase in the proportion of DRAQ7+ dead cells within the tissue (Fig. 2A, B, C). In contrast, mini-bladders exposed to low solute SHU showed an increase in the total number of cells and the proportion of EdU+ proliferating cells. (Fig. 2D, Supplementary Fig 4D, E).

Next, we explored the long-term consequences of urine solute concentration on tissue homoeostasis through transcriptional profiling of differentiated mini-bladder tissues exposed to high and low solute SHU for 8 days, using bulk RNAseq and subsequent gene set enrichment analyses (Fig. 2A, Supplementary Fig 5A). Long-term exposure to high solute SHU led to lower expression of cell-cell adhesion and proliferation gene sets (Fig. 2E), whereas gene sets related to developmental processes, particularly focusing on growth and differentiation, were enriched in samples exposed to low solute SHU (Supplementary Fig 5B), consistent with the morphological features noted above. In particular, genes related to urothelial differentiation, such as *KRT8, KRT5, KRT20, KRT13* and *UPK2* had lower expression in conditions with long-term exposure to high solute SHU (Fig. 2F), also reflected in qRT-PCR data (Supplementary Fig 5C). Genes related to intercellular junctions, such as *TJP2, TJP3, CLDN1, CLDN2* and *CDH2* were also downregulated by long-term exposure to high solute SHU (Fig. 2F). This transcriptional signature is corroborated by reduced ZO-1 immunostaining and higher permeability in the dextran diffusion assay (Fig. 2G, H). Interestingly, under high solute conditions, there was also a broad suppression of innate immune responses that was evident in the negative enrichment score of several gene sets for immune signalling and responses to bacterial infections (Fig. 2E, Supplementary Fig 5D). Notable amongst these are genes for toll-like receptor *TLR4, CASP1, NF-κB1*, cytokines *IL-6*, and chemokines *CXCL1* and *CXCL2* (Fig. 2I).

Taken together, these data show that long-term exposure to high solute urine impacts the resilience of the bladder tissue by weakening tight junctions, reducing stratified urothelial differentiation, and suppressing baseline immune responses.

### Urine with high solute concentration increases UPEC invasion and lowers antibiotic clearance in the tissue

Based on the above findings, we postulated that concentrated urine could predispose the bladder towards an increased susceptibility to infection. The mini-bladder is an ideal system to test this, as UPEC pathogenesis in urine and tissue can be studied simultaneously in a way that was not previously possible. First, we established a baseline protocol for infection in PHU with a mid-range of solute concentration

(~427 mOsm/kg) (Supplementary Table 1) and characterised the infection outcomes with CFT073 and UTI89, two widely used UPEC strains. Mini-bladders were inoculated with ca. 5 × 10$^7$ (CFT073) or 1 × 10$^7$ (UTI89) bacteria constitutively expressing GFP in PHU in the lumen (Supplementary Fig 6A, B). This corresponded to an approximate multiplicity of infection (MOI) = 50 for CFT073 (MOI = 10 for UTI89). The urinal population was sampled before and after a 10-h infection period, and the total bacterial load (within the urine and tissue) were quantified by fluorescence imaging (Supplementary Fig 6C, D). Both urinary and tissue bacterial burdens showed a steady increase over time, and characteristic features of UPEC infections such as IBCs and filamentous bacteria were visible at 10 h post-infection (hpi) (Supplementary Fig 6E, F). Similar 6-h infection periods in high and low solute SHU showed filamentous UPEC in the effluent—confirming the suitability of SHU formulation as a urine mimic (Supplementary Fig 6G).

Untreated chronic infections in humans involve a rapid turnover of bacteria with each micturition cycle, which previously could not be modelled in vitro. We can simulate this pathology in the mini-bladder by washing the lumen of the mini-bladder periodically to mimic micturition (Fig. 3A, B). This also provides a measurement of urinary bacterial load by collecting and plating for CFU. The urinary bacterial load reduced at least 10-fold after each wash (Fig. 3C), but washing did not completely clear the lumen. This is not unexpected as bacteria near the epithelial layer boundary experience lower urine flow rates[59], and may also adhere better to the umbrella cells because of the catch-bond mechanism of type I pili adhesion. In each wash, we detected exfoliated epithelial cells harbouring bacterial clusters in the collected urine (Supplementary Fig 6H). The urinary population recovered to a consistent peak of ca. 5 × 10$^8$ CFU/mL in the intervening growth phases, a pattern consistent with a chronic, untreated infection[21]. By optical microscopy, we identified a population of bacteria that co-localise with the tissue after removing the urinary bacterial population by a wash cycle (hereafter referred to as tissue-associated bacteria). This population of adhered and invaded bacteria increased steadily over several micturition cycles (Fig. 3D). UPEC adheres to umbrella cells via molecular interactions between the type I pili and the mannose units of the surface protein UP1A. This interaction can be competitively inhibited by the polysaccharide D-(+)-mannose[60–62]. The addition of D-mannose to the initial inoculation and the sterile PHU used for subsequent washes significantly reduced the tissue-associated bacterial load, verifying that this is a population that is either attached to or invaded in the tissue (Fig. 3B, D). Interestingly, the inhibition of UPEC adhesion (a pre-requisite to bacterial invasion) also reduced the urinary bacterial load after multiple micturition cycles (Fig. 3C), demonstrating that the tissue-associated bacteria contribute to reseeding the urinary bacterial population after micturition. When the tissue is infected under similar micturition cycles with CFT073 Δ*fimH* mutant, the total population of tissue-associated bacteria and the urinal bacterial load at 9.5 hpi are comparable to infection with D-mannose (Fig. 3E). D- mannose administration also reduced loss of tissue permeability post infection and tissue death (Fig. 3F-I), confirming a role for tissue-associated bacteria in driving these phenotypes. The human mini-bladder model, therefore, retains viability over several micturition cycles and uniquely recapitulates the dynamics of crosstalk between urine and tissue-associated UPEC in infections.

Next, we evaluated the role of urine solute concentration as a variable in driving favourable or deleterious outcomes in infection and therapy. We infected otherwise identical mini-bladders after 5 days of exposure to high and low solute SHU, treated with Ciprofloxacin at 100 X MIC (5 µg/mL), and observed the regrowth post-treatment (Supplementary Fig 7A, B). Prior to treatment at 3hpi, there was no difference in the urinary bacterial burden (Supplementary Fig 7C), consistent with the nearly identical doubling time of UPEC cultured axenically in these two urine compositions (Supplementary Table 1). However, at 3 hpi,

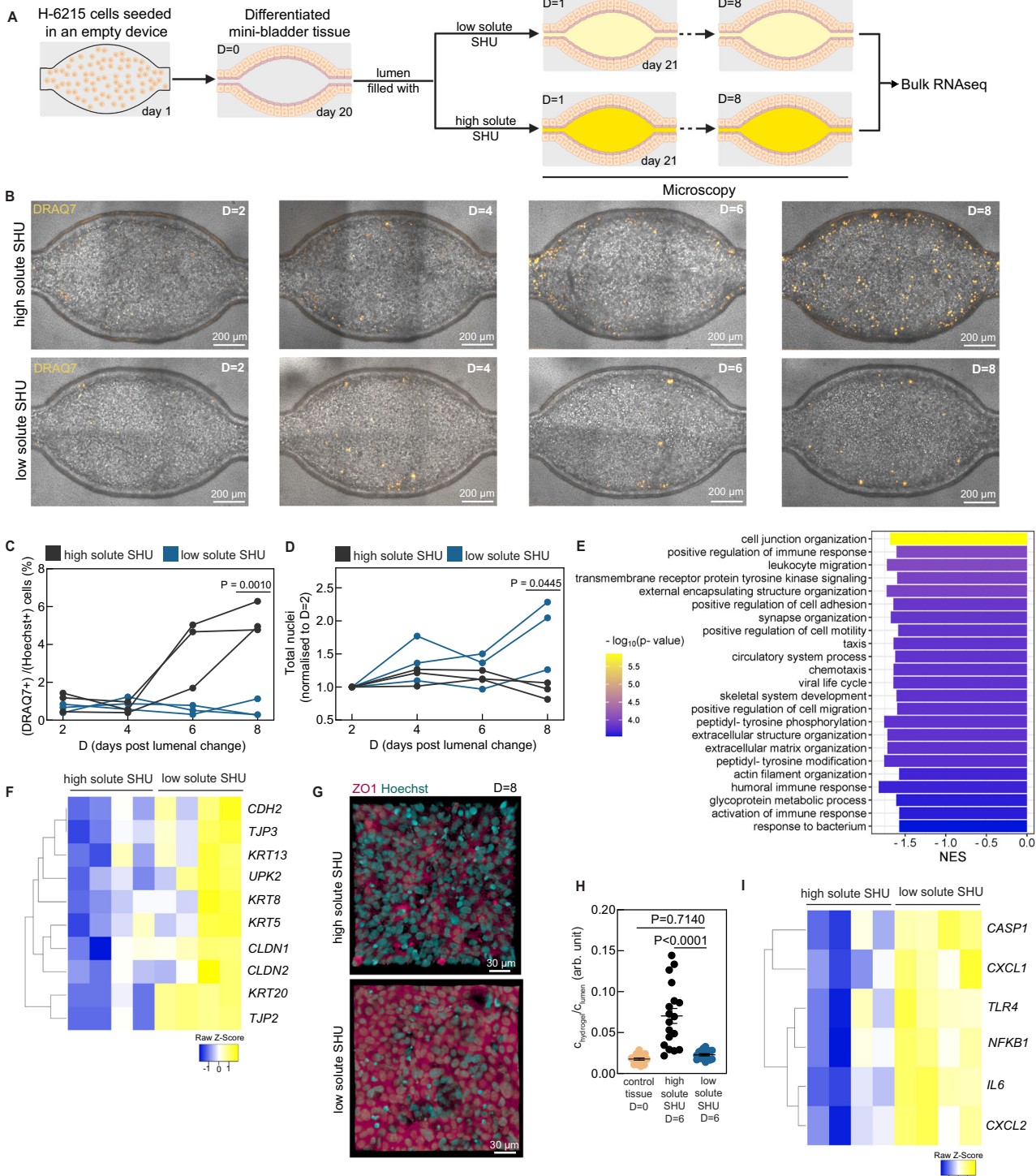

**Fig. 2 | High solute concentration in urine is detrimental to bladder tissue resilience.** A Timeline of experiments for long-term exposure of differentiated mini-bladders ($D = 0$) to low and high solute SHU for $D = 8$ days (created using BioRender. McKinney, J. (2026) https://BioRender.com/55huf39). **B** Snapshots (maximum intensity projection) from time-lapse imaging of mini-bladders in high and low solute SHU (DRAQ7, amber). Frequency of dead cells (DRAQ7+ nuclei, **C**) and total cell numbers (Hoechst+ nuclei, **D**) over time for $n = 3$ mini-bladders in high and low solute SHU (**E**) Top negatively enriched GO BP pathways ordered by statistical significance in high solute SHU exposed tissue at $D = 8$ compared to low solute SHU exposed tissue at $D = 8$ (FDR cut off <0.1%); Normalised enrichment score (NES). **F** Selected genes ($p < 0.05$) related to intracellular junction and tissue differentiation compared between high solute SHU-exposed tissue and low solute-

exposed tissue. **G** Representative maximum intensity projection images of ZO-1 signal in high solute SHU exposed tissue at $D = 8$ compared to low solute SHU exposed tissue at $D = 8$ (ZO-1, magenta; Hoechst, cyan) (**H**) Quantification of relative dextran permeability between high solute SHU exposed tissue at $D = 6$, low solute SHU exposed tissue at $D = 6$ and control tissue at $D = 0$ ($n = 3$ biological replicates each) (**I**) Selected genes ($p < 0.05$) related to immune signalling compared between high solute SHU exposed tissue and low solute exposed tissue. Data represented as mean ± SEM. *P*-values calculated using a two-sided unpaired t-test in (**C**, **D**), two-sided false discovery rate (FDR) correction applied for multiple comparisons in (**E**), one-way Anova with Dunnett's multiple comparison test in (**H**). Image in (**G**) is representative of at least 3 biologically independent experiments. Source data are provided as a Source data file.

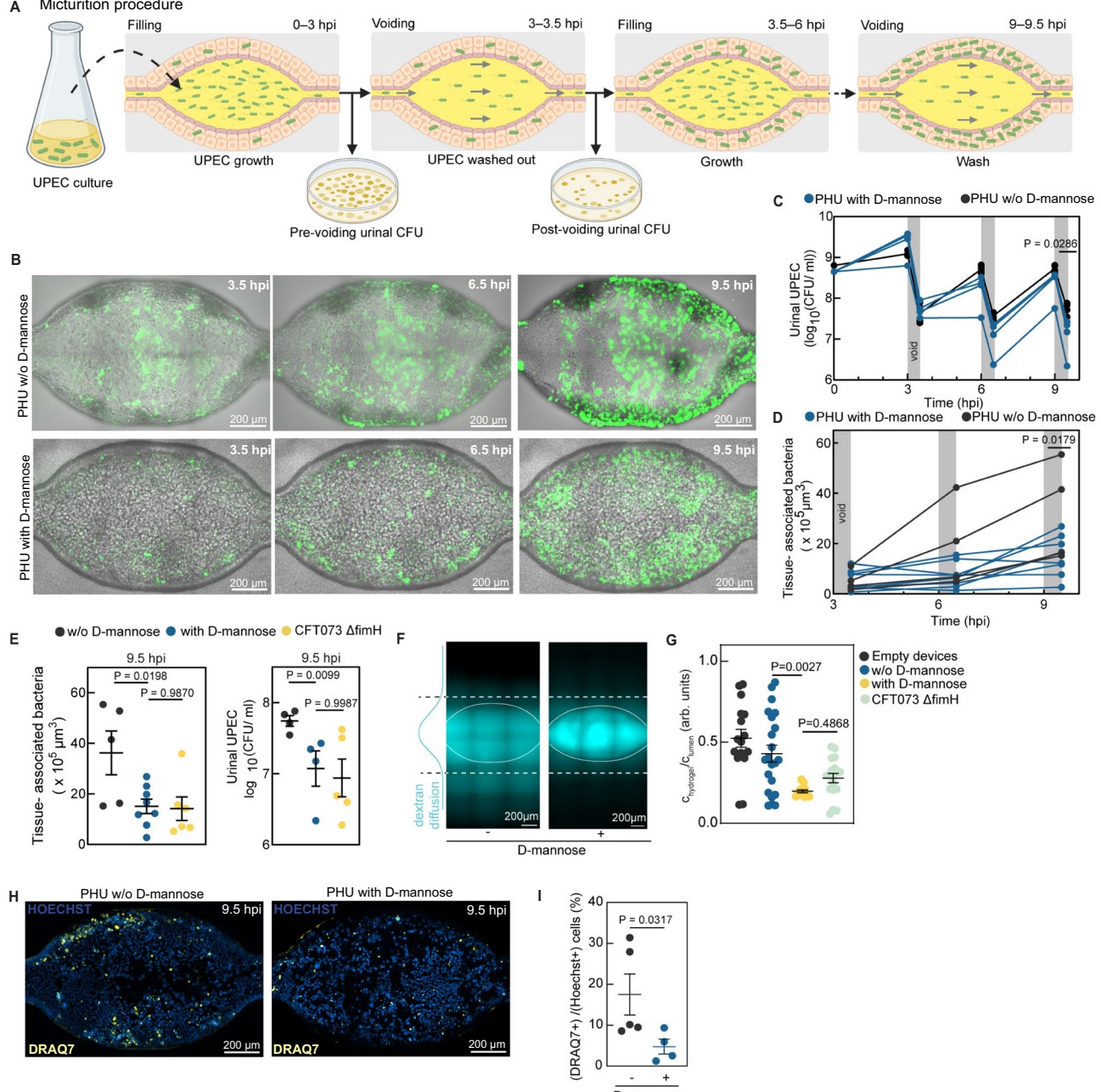

**Fig. 3 | A dynamic crosstalk between urine and urothelial UPEC populations in chronic UTIs. A** Schematic of UPEC infection with micturition cycle. The lumen is washed with sterile urine every 3 h (created using BioRender. McKinney, J. (2026) https://BioRender.com/55huf39). **B** Snapshots (maximum intensity projection) of timelapse imaging of UPEC-infected mini-bladders with and without 11.1 mM D-mannose added in pooled human urine (PHU) (UPEC, green). **C** Urinal CFU collected before and after periodic washes from ($n = 4$ each) mini-bladders with and without D-mannose. **D** Tissue associated bacterial volume measured after periodic washes from mini-bladders with ($n = 8$) and without ($n = 4$) D-mannose. **E** Tissue associated bacterial volume and urinal CFU measured at 9.5 hpi post periodic washes from mini bladders with D-mannose ($n = 8$; 4 samples respectively), without D-mannose ($n = 5$; 4 samples respectively) and infected with CFT073 Δ*fimH* ($n = 6$;

5 samples respectively). **F** Representative images from the dextran diffusion assay at $t = 9.5$ hpi (quantified in (**G**)). (**G**) Quantification of dextran permeability from devices with D-mannose, without D-mannose, infected with CFT073 Δ*fimH* and empty devices ($n = 2, 4, 3, 3$ biological replicates each). **H** Representative images (maximum intensity projection) of mini-bladders at 9.5hpi with and without D-mannose (Hoechst, azure; DRAQ7, yellow) (quantified in (**I**)). **I** Frequency of dead cells (DRAQ7+ nuclei) at 9.5 hpi from mini-bladders with ($n = 4$) and without ($n = 5$) D-mannose. Data represented as mean ± SEM. *P*-values calculated using two-sided unpaired t-test in (**D**, **I**), two-sided Mann–Whitney in (**C**), one-way Anova with Dunnett's multiple comparison test in (**E**, **G**). Source data are provided as a Source data file.

the tissue-associated bacterial burden was significantly higher under high solute concentration conditions, which can be explained by improved access to deeper layers of the tissue caused by the loss of tight junctions and differentiated cells from long-term exposure to high solute SHU (Supplementary Fig 7D). Subsequent treatment with

Ciprofloxacin, an antibiotic effective even against slow-growing bacteria, showed a stark difference in treatment efficacy. Clearance of urinal bacteria was independent of SHU solute concentration, whereas tissue-associated bacteria exhibit increased survival in mini-bladders with long-term exposure to high-solute urine (at 9.5 hpi)

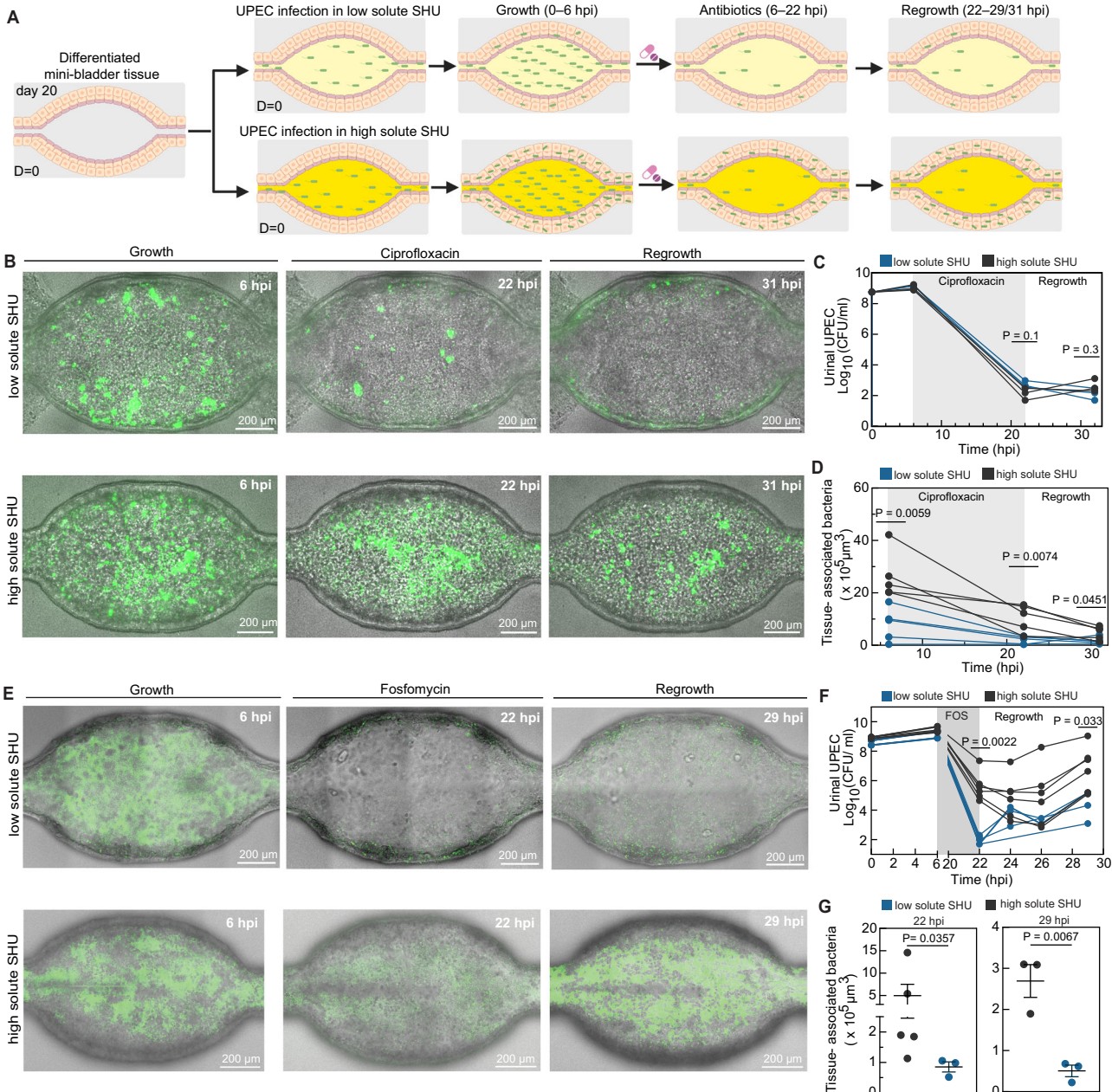

**Fig. 4 | High solute concentration in urine lowers antibiotic clearance in the tissue. A** Schematic representation of the timeline of UPEC infection and treatment (created using BioRender. McKinney, J. (2026) https://BioRender.com/55huf39). **B** Snapshots (maximum intensity projection) of timelapse imaging of mini-bladders infected and treated with Ciprofloxacin (100X MIC in high solute SHU, 5 µg/mL) in high and low solute SHU (UPEC, green). **C** Urinal CFU collected through washes from mini-bladders ($n$ = 3 samples each) in high and low solute SHU. **D** Tissue-associated bacterial volume measured from mini-bladders with high ($n$ = 5 samples) and low ($n$ = 5 samples) solute SHU. **E** Snapshots (maximum intensity projection) of

time-lapse imaging of mini-bladders infected and treated with Fosfomycin (40X MIC in high solute SHU, 300 µg/mL) in high and low solute SHU (UPEC, green). **F** Urinal CFU collected through washes from mini-bladders ($n$ = 6 samples each) in high and low solute SHU. **G** Tissue associated bacterial volume measured from mini-bladders with high ($n$ = 5 samples) and low ($n$ = 3 samples) solute SHU at 22 hpi and from mini-bladders ($n$ = 3 samples each) with high and low solute SHU at 29 hpi, post fixation. Data represented as mean ± SEM. *P*-values calculated using two-sided Mann–Whitney in (**C**, **F**, **G**) at 22 hpi; two-sided unpaired t test in (**D**, **G**) at 29 hpi. Source data are provided as a Source data file.

(Supplementary Fig 7C, D). To disentangle if this effect is due to the lack of tissue resilience or a differential bacterial response to antibiotics based on spatial niche of UPEC survivors, or both, we also performed similar experiments without the 5-day high/low solute SHU exposure, with longer infection and treatment periods (Fig. 4A). At $D = 0$, the tissue associated bacteria at 6hpi is lower under low solute concentration conditions in comparison to high solute concentration and the difference is negated when the tissue has been pre-treated with EDTA to disrupt tight junctions (Supplementary Fig 7E). Interestingly,

the protective trend against Ciprofloxacin for the tissue-associated bacteria but not the urinal bacteria was also evident even during this longer infection and treatment cycle (Fig. 4B, C, D). When normalised for bacterial burden at the start of antibiotic treatment, there is no difference in the clearance rate or recovery of tissue-associated UPEC after antibiotic treatment Supplementary Fig 7F, reinforcing the higher tissue-associated bacterial load accumulated prior to ciprofloxacin treatment as a driver for the differences in Fig. 4B–D. Unlike ciprofloxacin, experiments with Fosfomycin (FOS) administration (40X

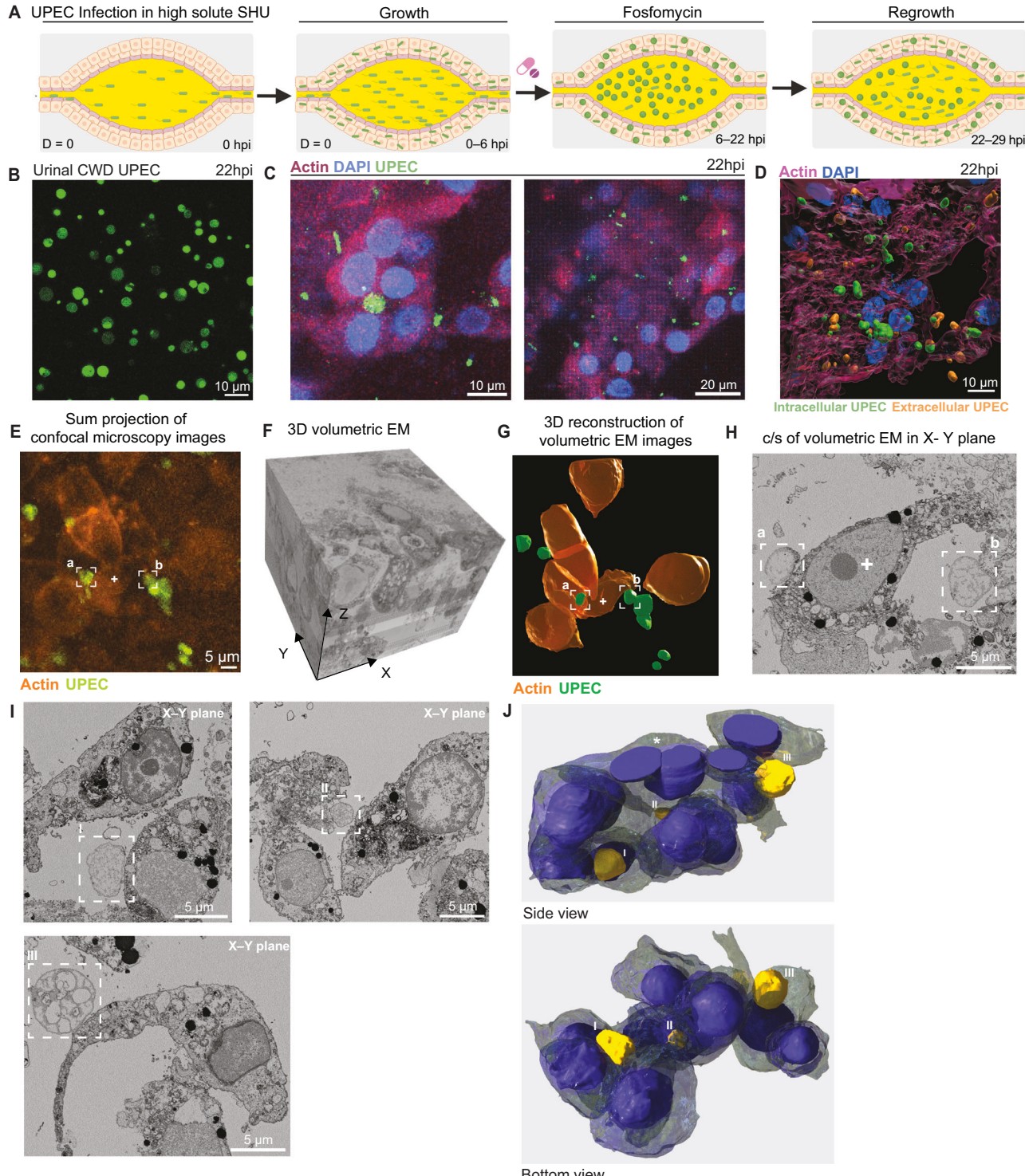

A UPEC Infection in high solute SHU | Growth | Fosfomycin | Regrowth
D = 0  0 hpi | D = 0  0–6 hpi | 6–22 hpi | 22–29 hpi

B Urinal CWD UPEC  22hpi

C Actin DAPI UPEC  22hpi

D Actin DAPI  22hpi
Intracellular UPEC  Extracellular UPEC

E Sum projection of confocal microscopy images
Actin UPEC

F 3D volumetric EM

G 3D reconstruction of volumetric EM images
Actin UPEC

H c/s of volumetric EM in X- Y plane

I X–Y plane

J Side view

Bottom view

MIC, 300 µg/mL) showed not only tissue-associated bacteria but also urinary bacteria were protected from clearance in high solute concentrations (Fig. 4E, F, G). Taken together, these experiments with two antibiotics with different modes of action demonstrate that urine solute concentration can impact therapeutic outcomes through two mechanisms. First, higher solute concentration reduces tissue resilience and enables higher colonisation of the tissue by bacteria. In turn, bacteria within the tissue are further protected from clearance by antibiotics. In the case of FOS, this protection also extends to the urinal bacteria.

## Tissue-invaded UPEC under urine with high solute concentration can survive in cell wall deficient form during Fosfomycin treatment

Next, we examined the effects of solute concentration on Fosfomycin (FOS) therapy in greater detail as FOS, a cell-wall targeting antibiotic, is routinely prescribed for uncomplicated UTIs and administered as a single dose that accumulates in the urine[63–65] (Fig. 5A). The inefficacy of FOS in reducing urinal UPEC load in high solute SHU was clearly evident at 22 hpi (Fig. 4F) and resulted in faster regrowth and higher bacterial burdens at 29 hpi (Fig. 4G). An examination of the sampled

**Fig. 5 | Tissue associated UPEC can survive in cell wall deficient forms after Fosfomycin treatment in urine with high solute concentration. A** Schematic representation of the timeline of UPEC infection and treatment with Fosfomycin (40 X MIC in high-solute SHU, 300 μg/mL) in high-solute SHU (created using BioRender. McKinney, J. (2026) https://BioRender.com/55huf39). **B** Representative image of urinal bacteria in high solute SHU at 22 hpi (UPEC, green). **C** Representative cross sections of high solute SHU exposed tissue at 22 hpi after Fosfomycin treatment (actin, magenta; DAPI, azure; UPEC, green). **D** Representative 3D view of Fosfomycin-treated tissue at 22 hpi (actin, magenta; DAPI, azure; Intracellular UPEC, green; Extracellular UPEC, amber). (**E**) Sum intensity projection of confocal images along the Z plane of Fosfomycin-treated tissue at 22 hpi (amber, epithelial cells; green, UPEC). Representative CWD UPEC (marked in white dashed boxes) and epithelial cell (marked in white plus sign) to compare across (**E, G, H**). **F** Volumetric EM image stack of Fosfomycin treated tissue section shown in (**E**). **G** 3D reconstruction of volumetric EM stack shown in (**F**) (amber, epithelial cells; green, UPEC). Representative CWD UPEC (marked in white dashed boxes) and epithelial cell (marked in white plus sign) to compare across (**E, G, H**). **H** Representative cross-sectional image in X-Y plane of (**F**). Representative CWD UPEC (marked in white dashed boxes) and epithelial cell (marked in white plus sign) to compare across (**E, G, H**). **I** Representative cross-sectional images in the X-Y plane from volumetric EM of Fosfomycin treated tissue at 22 hpi (CWD UPEC, marked in white dashed boxes). **J** 3D reconstruction of the volumetric EM images (represented in (**I**)) shows surviving cell wall-deficient UPEC (yellow) in between epithelial cells (nuclei, purple; cell membrane, grey). CWD UPEC numbered corresponding to (**I**); Bi nucleated umbrella cell marked by a white asterisk. Images in (**B–E, H, I**) are representative of at least 3 biologically independent experiments.

urine at 22 hpi by confocal microscopy showed the presence of large, GFP+ forms, with a high degree of sphericity (Fig. 5B), very different from the relatively fewer (Fig. 4F) and predominantly rod-shaped bacilli obtained in low solute SHU (Supplementary Fig 8A, B, C). These large spherical morphologies are consistent with the reported structure of CWD or L-form UPEC[31,32,66]. CWD UPEC have been identified in the urine of human patients with chronic infections and as a response to FOS treatment in infections in zebrafish larvae[32]. We performed several tests to verify whether this was the case in our human model. Mickiewicz et al.[32] showed that CWD forms are sensitive to hypoosmotic stress and show a defect in recovery on standard LB agar plates, which is abrogated by plating on osmoprotectant LM agar plates (containing 1x brain heart infusion and 0.58 M + D-sucrose). Exponential phase UPEC cultured axenically in LB shows no difference in recoverable CFU on both formulations (Supplementary Fig 8D). The culturable urinal bacterial load at 22 hpi plated on LM plates was consistently higher than recovery on LB plates for high solute SHU in comparison to low solute SHU, consistent with our assignment of the spherical GFP+ forms as CWD UPEC (Supplementary Fig 8E, F). This difference is negated post regrowth at 29 hpi, suggesting that the CWD UPEC converted back to rod-shape once the antibiotic pressure is removed (Supplementary Fig 8G, H). As further proof, we generated CWD UPEC in axenic culture in high-solute SHU using the same treatment regimen, and imaged these structures using transmission electron microscopy (TEM) (Supplementary Fig 8I). After 16 h of FOS treatment, although a vast majority of bacteria are CWD, we also observed some bacteria in transition from rod-shaped to CWD (Supplementary Fig 8J). The size and morphology of the CWD UPEC in the TEM images are comparable to those obtained by confocal imaging (Supplementary Fig 8K). CWD UPEC were also observed axenically in high solute PHU upon similar treatment with Fosfomycin (Supplementary Fig 8L).

Intriguingly, high-resolution confocal microscopy imaging of fixed mini-bladder tissue revealed a heterogenous population of rod-shaped and spherical bacteria within the tissue (Fig. 5C). The latter population had cell sizes consistent with the morphology of CWD UPEC in the urine, and not the densely-packed smaller coccoid bacteria observed in IBCs[30,40]. The spatial localisation of the GFP signal overlapped significantly with that of actin, showing that these bacteria were within the tissue in either an intracellular or pericellular association (Fig. 5C, D). The exact localisation could not be resolved with the optical resolution of confocal microscopy. To examine this interaction with improved resolution, we performed correlated optical microscopy and volumetric EM on the FOS- treated tissue at 22 hpi. Potential CWD forms were identified as GFP+ via confocal imaging (Fig. 5E). Excitingly, when comparing the overall spatial orientation between the confocal and electron microscopy images, many of these GFP+ forms mapped to similar spherical structures in the 3D reconstruction of volumetric EM images (Fig. 5E, F, G). These spherical structures also had morphologies and contrast profiles consistent with axenically generated CWD UPEC imaged with the TEM (Fig. 5H, Supplementary Fig. 8I). These 'tissue-associated CWD UPECs' were observed attached to epithelial cells in a pericellular manner at multiple independent spatial locations throughout the tissue; representative cross-sections are shown (Fig. 5I). Detailed 3D reconstruction showed a CWD UPEC residing within a crevice beneath a double-nucleated umbrella cell and positioned between two other adjacent cells (Fig. 5J). Additionally, CWD UPEC was also identified between cells deeper within the tissue away from the umbrella layer, coincident with the intermediate or basal cell population described in Fig. 1E. Our results show that CWD are not solely a urinal phenomenon, and that bacteria within the tissue can also adopt CWD forms.

Next, we wondered where there is also an interplay between tissue-associated and urinal bacteria in the case of FOS treatment and CWD UPEC formation in the bladder. To this end, we examined if the presence of urinary CWD UPEC alone was sufficient to account for the overall higher burdens in mini-bladders 7 h after removal of antibiotic therapy. To achieve this, we treated UPEC in axenic cultures of high and low solute SHU with FOS and reseeded the survivors into an otherwise uninfected mini-bladder lumen (Supplementary Fig. 9A). The survivors in high solute SHU were CWD forms of similar size and morphologies as previously observed (Supplementary Fig. 9B, C, Fig. 5B). Although the FOS treatment was far more effective in low solute conditions, and therefore far fewer bacteria were seeded into the mini-bladders, this small surviving population rapidly proliferated in the lumen to reach a burden of $10^6$–$10^7$ CFU/mL after 7 h (Supplementary Fig. 9C, D). In contrast, although the proportion of survivors was much higher in high solute SHU, these were primarily CWDs (Supplementary Fig. 9B, C), and proliferated much slower over the course of 7 h (Supplementary Fig. 9D). This is likely due to the difference in division rate of CWD UPEC compared to rod-shaped bacteria, or a significant lag period during which CWD reconverted to rod-shaped bacteria before proliferation[31,67]. Although more survivors were introduced in the mini-bladder in high solute SHU at 22 hpi, the final tissue-associated bacterial load post 7 h of regrowth was not significantly different in both cases (Supplementary Fig. 9E). However, this is not the case when CWD forms are generated in the mini-bladder (Fig. 4G). This demonstrates that without tissue-associated niches formed by CWD UPEC, there is no significant difference in the regrowth and recurrence between high and low solute SHU after FOS treatment, highlighting the crucial contribution of intercellular CWD UPEC survivors in regrowth kinetics.

These results collectively expand the survival niches of CWD UPEC (previously considered solely a urinary population solely) to the tissue and demonstrate the role of urine solute concentration in driving the population of this niche, underscoring its critical role in recurrence. Furthermore, this highlights the significance of the mini-bladder model in capturing the dynamic interactions of different bacterial subpopulations within urine and urothelium, enhancing our understanding of recurrence after antibiotic treatment.

## Discussion

The development of an in vitro model that accurately replicates the multicellular, structural and mechano-dynamic complexity of the bladder has long been a challenge. The mini-bladder model addresses several previously unattainable requirements in one model. It has a stratified, well-differentiated urothelium that can be exposed to urine for several days, with micturition cycles and mechanical stretch and flow, thus effectively combining the strengths of bladder organoid, bladder-on-chip and Transwell models reported previously[30,40,41]. These functionalities allow us to explore the role of urine as an experimental variable in tissue homoeostasis and infection for the first time. This is challenging to achieve on a consistent basis, even in animal models, where urine composition is well-controlled for welfare reasons and does not capture the diversity observed in humans. Interestingly, we find that consistent exposure to high-solute urine is detrimental to tissue health, confirming reports in vivo in animal models[13–16,43]. Our results suggest that this reduces the resilience of the bladder to infection, which may be an overlooked factor in populations prone to urinary dysbiosis, such as those with deficient renal function or patients with catheters.

The mini-bladder allows us to examine the immediate consequences of variable urine composition with high spatiotemporal resolution. The model can recreate a degree of chronicity by the incorporation of micturition-mimicking washes. This extends the time course of analysis beyond the typical one cycle of growth that is possible in organoid-based approaches and takes in vitro models closer to the in vivo bladder with a high turnover of urinary bacteria and replenishment of nutrition. Our work confirms a dynamic crosstalk between tissue-associated bacteria and the urinary bacteria within hours of infection. We go on to show that the loss of bladder resilience under urine with high-solute concentration conditions accelerates bacterial invasion and generates higher bacterial burdens in tissue. Co-morbidities that alter urine composition can have a detrimental synergistic effect in infection and generate favourable conditions for a relapse of infection.

The role of urine as a variable in tuning the balance between the urinary and tissue-associated bacterial populations becomes all the more pertinent in the context of therapy. Between two antibiotics with different mechanisms of action, we consistently observe delayed clearance of tissue-associated bacteria, independent of but likely exacerbated by the loss of bladder resilience. Specifically in the case of FOS, this also extends to the urinary bacterial load. These findings are pertinent given the widespread use of FOS in uncomplicated infections[63].

To the best of our knowledge, this is the first in vitro tissue model that can recapitulate CWD formation, allowing us to examine the dynamic role of this phenomenon in the overall infection paradigm in a manner that cannot be achieved with human patients. Our work expands the conceptual understanding of the phenomenon of CWD formation to show that they can also form and exist within tissue, and are a contributory factor to the urinal and tissue bacterial burden after antibiotic removal. Future work will examine the mechanisms by which tissue-associated CWD UPEC form and whether there are unique signatures that can distinguish them from CWD forms in the urine, and the mechanisms by which they recover and regrow. Beyond the clear relevance for antibiotic therapy, these findings have important implications for bladder immunology, as CWD bacteria may be able to evade several immune-sensing mechanisms[68].

In conclusion, the mini-bladder model provides a unique platform for a detailed exploration of the tripartite interaction between urine composition, the urothelium, and UPEC.

Despite the advances in this report, some limitations in the full recapitulation of the 3D stratification of a human bladder remain. These include a variability in the number of cell layers and umbrella cell size, the absence of widespread AUM plaques and uroplakin staining uniformly lining the lumen, and the relatively higher cellular proliferation rate. This may be linked to the absence of vasculature, immune and mesenchymal components, which contribute to tissue homoeostasis. These will be the focus of further model development. Our work also reveals the limitations of the SHU formulation in understanding the role of specific urine components on bladder physiology, highlighting the development of synthetic human urine formulations that better captures the solute concentrations across different hydration states as a focus for future efforts.

## Methods

This study complies with the ethical regulations of Commission cantonale d'Éthique de la Recherche sur l'être humain Vaud (CER-VD)).

### Microdevice design and fabrication

The microfluidic device was fabricated as previously described in Nikolaev. M et al.[44] using conventional soft lithography and poly-dimethylsiloxane (PDMS) moulding. Briefly, the device has three main compartments: a hydrogel compartment for tissue culture in the centre, two larger open (basal) media reservoirs flanking the hydrogel compartment, and a pair of smaller inlet/outlet reservoirs (Supplementary Fig. 1A). The liquid hydrogel precursor was loaded through an additional inlet (hydrogel port) and then polymerised (details below). Medium from the two basal side reservoirs can passively diffuse into the hydrogel. The chips were UV sterilised and kept sterile until further use.

### Hydrogel loading and microchannel fabrication

Extracellular matrix (ECM) solution containing neutralised 6 mg/mL bovine TeloCol®–6 type I collagen solution (Advanced Biomatrix) was injected into the hydrogel compartment of the microdevice through the hydrogel loading port and incubated at 37 °C for 10 min. Thereafter, the cell loading ports and media reservoirs were filled with PBS. Laser ablation of the hydrogel was performed using a customised two-photon laser microscope co-developed with UpNano GmBH and equipped with a 10×/0.25NA objective (Zeiss), 800 nm pulsed 1.5 W laser. To generate the mini-bladder form, a 3D model of the bladder was created in Autodesk Fusion 2025, and the mesh was generated using Blender 4.5. The resulting model was imported into the microscope control software and positioned along the hydrogel compartment of the microdevice, covering its entire length, at $Z = 100\,\mu m$ from the bottom glass coverslip. After laser ablation, the channel was perfused with PBS and then stored in the fridge (4 °C) until use for up to several weeks. Overnight before use, PBS was exchanged to H6621 complete epithelial cell medium (Cell Biologics) and then maintained in the incubator (37 °C, 5% CO₂).

### Cell culture of human bladder epithelial cells

H-6215 primary human bladder epithelial cell line (Cell Biologics) obtained from female donors was cultured in H6621 complete epithelial cell medium (with supplied kit) supplemented with 10% Heat inactivated Foetal Bovine Serum (HI- FBS) (henceforth called H6621 complete media) as recommended by the supplier. Epithelial cells were passaged by detachment with TryPLE Express Enzyme (1×) at 37 °C for 3–5 min followed by neutralisation of TryPLE with H6621 complete medium. The cells used in all the experiments were at ten passages or fewer. The cells tested negative routinely for mycoplasma contamination during passaging.

### Pooled and synthetic human urine

Urine was collected from 10 consenting individuals of various sex at various hours of the day, and osmolarity was measured using a micro-osmometer (Osmo1® Single-Sample Micro-Osmometer; Advanced Instruments) (BASEC ID–2025-01193 under Commission cantonale d'Éthique de la Recherche sur l'être humain Vaud (CER-VD)). The

4 samples with the highest osmolarity were pooled to obtain the PHU with high solute concentration. Similarly, the 4 samples with the lowest osmolarity were pooled to obtain PHU with low solute concentration. PHU with mid-range solute concentration was procured from BioIVT, UK (filter-sterilised human urine, mixed sex). All PHU samples were filtered using Stericup Vacuum Filters (Merck, pore size 0.22 μm) and stored at −20 °C in aliquots of 5 mL.

Synthetic human urine was fabricated using the protocol from Ipe et al.[56]. This corresponds to high solute SHU. The detailed composition is listed in Supplementary Table 2. All components were stirred until complete dissolution, and the pH was adjusted to 5.6 with 1 M NaOH or 37% HCl for long-term stability, as neutral pH often resulted in precipitation of components when stored at 4 °C. The solution was filtered using Stericup Vacuum Filters (Merck, pore size 0.22 μm). Low solute SHU was made by diluting high solute SHU 5× in ddH$_2$O. The Iron (II) sulphate heptahydrate (FESO$_4$.7H$_2$O) and casamino acid concentrations were adjusted back in low solute SHU to the amount equal to high solute SHU. The pH was again adjusted to 5.6 with 1 M NaOH or 37% HCl. The SHU solutions were stored at 4 °C for a maximum of 1–2 months. Osmolarity was regularly measured using a micro-osmometer (Osmo1® Single-Sample Micro-Osmometer; Advanced Instruments). Temporary buffered aliquots with additional 20 mM HEPES and 1 mM CaCl$_2$ were made for both high and low solute SHU, this was used for all experiments related to differentiation and infection.

## Mini bladder tissue culture

To seed cells into the central channel of the etched mini-bladder, H-6215 cells (between passage numbers 4 and 10) were dissociated from T75 flasks with TryPLE Express solution for 5–10 min at 37 °C. The cells were washed with H6621 complete media and centrifuged at 300 g for 5 min. This process was repeated twice to remove residual TryPLE. After centrifugation, the pellet was resuspended in H6621 complete medium at a density of about 10$^8$ cells/mL. Media was removed from the microchannel inlet, outlet and basal side medium reservoirs, and 10 μL of cell suspension was allowed to fill the laser-ablated microchannel by gravity-driven flow. The cells were allowed to adhere for about 1–1.5 h, followed by gentle washes through the inlet and outlet to remove all non-adherent cells. The urinal and basal channels were filled with H6621 complete media, and the device was incubated at 37 °C in 5% CO$_2$ humidified air. The cells are grown with H6621 complete media in the urinal and basal chambers for 15 days to allow for tissue stratification. From day 16, the urinal channel was replaced with high solute SHU while the basal chamber was kept under H6621 complete media. The tissue was grown with urinal high solute SHU exposure for a further 5–7 days to complete the process of stratification and differentiation. During this period, the urinal and basal medium were changed every day. Tissue defects and failure of proper mini-bladder development, detected in around 10% of cases, may be caused by technical problems related to device preparation (e.g., defective microfluidic chips, hydrogel preparation) or issues related to cell culture. These defective tissues were not used for any experiments or analysis.

## Mini-bladder tissue stretching

The stretching experiments were performed using the CETONI Nemesys S syringe pump system with two modules. Two 1 mL syringes (Hamilton) were connected to both inlets of the chip. The first syringe was set to deliver a constant positive (dispensing) flow rate of 10 μL/min, while the second syringe was configured for variable negative (aspiration) flow rates. When both syringes were synchronised (inlet syringe: +10 μL/min, outlet syringe: −10 μL/min), a continuous steady-state perfusion was achieved. The inflation of bladder tissue was simulated by reducing or stopping the outlet syringe while maintaining a positive flow rate in the first syringe. A cyclic rectangular-wave

perfusion profile was programmed for the outlet syringe, varying the flow rate from 0 μL/min (urine accumulation, inflation phase) to −10 μL/min (bladder emptying, deflation phase). This setup enabled multiple cycles of automated perfusion with bladder stretching.

## Immunofluorescence staining

Mini-bladders were rinsed with PBS (Gibco) and fixed in 4% paraformaldehyde (PFA; Thermofisher) in PBS overnight at 4 °C. After rinsing with PBS (Gibco), the hydrogel compartment was cut around its perimeter using a razor blade to extract the block of hydrogel containing mini-bladder tissue. The samples were permeabilized with 0.1% Triton X-100 (Sigma-Aldrich) in PBS (Gibco) (30 min, RT) and blocked in 1% Bovine Serum Albumin in PBS (Gibco) (blocking buffer) at 4 °C for at least 8 h. Samples were subsequently incubated overnight at 4 °C with primary antibodies diluted in blocking buffer. The following primary antibodies were used: rabbit anti-human Cytokeratin 20 (1: 50, Biorbyt, orb256650), rabbit anti-human Uroplakin 3 A (1:50, Thermofisher, PA5-87581), rabbit anti-human Alexa Fluor 488-Cytokeratin 13 (1:50, Abcam, ab92551), rabbit anti-human Alexa Fluor 647-Cytokeratin 8 (1:50, Abcam, ab53280), goat anti-human ZO-1 (1:50, Abcam, ab190085), rabbit anti-human UP1A (1:50, Thermofisher, 11045-RBM9-P1ABX) diluted in blocking buffer. The samples were then incubated in blocking buffer for at least 5 h to wash unbound primary antibodies, then incubated overnight at 4 °C with secondary antibodies. The following secondary antibodies were used: Alexa Fluor 488 donkey anti-goat (1:1000, Thermofisher, A-11055), Alexa Fluor 700 goat anti-rabbit (1:1000, Thermofisher, A-21038) and Alexa Fluor 647 donkey anti-rabbit (1:1000, Thermofisher, A-31573) diluted in blocking buffer. The samples were stained for Hoechst solution 33342 (1: 300, Thermofisher, 62249), DAPI (1: 300, Thermofisher, 62248) or Alexa Fluor 555 - Phalloidin (1:400, Thermofisher, A34055) diluted in blocking buffer for 1 h at RT (based on the experiment).

## RNA extraction and quantitative real-time PCR (qRT-PCR)

The hydrogel block containing mini bladder tissue was extracted by cutting the hydrogel compartment using a razor blade. The hydrogel block was then incubated in Type I Collagenase (150 U/mL, Thermofisher, 17018029) diluted in PBS (Gibco) for 30 min at 37 °C. Additionally, the hydrogel was periodically sheared using a 1 mL pipette to accelerate disaggregation of the matrix. The sample was then centrifuged at 600 g for 10 min and the pellet was incubated in 700 μL of RNA lysis buffer with 0.14 M β-mercaptoethanol (β-ME) (RNAeasy Plus Micro Kit, QIAGEN). RNA was isolated following the manufacturer's instructions and resuspended in 14 μL of DEPC-treated water. 11 μL of the RNA-containing solution was used to generate cDNA using the SuperScript®IV First-Strand Synthesis System with random hexamers (Invitrogen), which was stored at −20 °C. qRT-PCR primer sequences are listed in Supplementary Table 3. qRT-PCR reactions were prepared with SYBRGreen PCR Master Mix (Applied Biosystems) with 500 nM primers, and 1 μL cDNA. Reactions were run for quantification on QuantStudio 6 or 7 Flex Real-Time PCR System (Thermofisher), and amplicon specificity was confirmed by melting-curve analysis.

## RNA-sequencing and data analysis

Libraries were prepared from ribodepleted RNA isolated from human bladder microtissue and sequenced using the AVITI sequencer (PE75). A mean of 53.3 million 75-base-pair paired-end reads was obtained among the 11 samples (43.23–75.74 million reads). The quality of sequenced Fastq files was analysed using FastQC (version 0.11.10), and reads were mapped on the human reference genome assembly hg38 using STAR (v2.7.10b)[69]. The uniquely mapped reads were between 82.6 and 94.3%. Gene counts were generated using FeatureCounts[70], and differential expression analysis was performed with the DESeq2 (v.1.42.1)[71] package from Bioconductor (v3.18)[72]. Genes were considered differentially expressed based on an adjusted p-value cutoff of

<0.05. Libraries normalisation was performed using the variance stabilising transformation method of DESeq2. This normalisation produces transformed data on a log (2) scale. The point of these transformations is to remove the dependence of the variance on the mean, particularly the high variance of the logarithm of count data when the mean is low. VST is very similar to the regularised logarithm or rlog. Gene Set Enrichment analysis was carried out using the GSEA[72] function in ClusterProfiler[73] (v4.10.1), and the following annotated gene sets from MSigDB v6.2[74]: the Hallmark gene set[75].

## UPEC infection of mini-bladder

The UPEC CFT073 strain was originally isolated from a patient with bacteraemia of urinary tract origin[76] and procured from ATCC. A derivative strain expressing green fluorescent protein (GFP) was generated by introducing the sfGFP sequence from plasmid pSLC293 under the control of a strong Pσ70 promoter at the chromosomal attHK022 site (between nucleotides 1,090,927 and 1,090,928) using a recently described method[77] (adapted from Simonet et al.[78]). The *fimH* locus (between nucleotides 5,143,529 and 5,144,440) was deleted from CFT073 sfGFP using the λ Red recombineering system (adapted from Simonet et al.[78]). The UPEC UTI89 strain, recovered from patients with acute cystitis, was gifted from the Guet lab at IST Austria. Similar to CFT073, a derivative strain expressing green fluorescent protein (GFP) was generated by introducing the sfGFP sequence from plasmid pSLC293 under the control of a strong Pσ70 promoter at the chromosomal attHK022 site (between nucleotides 1044461 and 1044640).

To induce expression of type 1 pili, UPEC was grown in LB media under non-shaking conditions at 37 °C overnight prior to the experiment, to achieve a stationary phase culture (for CFT073–$OD_{600} = 1.5$ and corresponding to a concentration of ~$1.5 \times 10^9$ bacteria/mL). The bacteria were diluted 3-fold (final concentration of ~$5 \times 10^8$ cells/mL for CFT073) by centrifuging (2500 g, 5 min) and resuspended in either pooled or high/low solute synthetic human urine (temporary aliquots made with additional 20 mM HEPES, 1 mM $CaCl_2$), based on the experiment. For all infection experiments, 100 µL of this suspension was introduced into the lumen through one reservoir. This was flowed through the channel, collected from the other reservoir and reintroduced into the original reservoir. After two such rounds, the mini-bladder was considered infected. For CFT073, this corresponded to an inoculum of ~$5 \times 10^7$ bacteria and an MOI of 1:50 (assuming that the superficial cell layer is populated by the same number of epithelial cells that initially form a monolayer in the mini-bladder, ca. $10^6$ cells). Similarly, for experiments with UTI89 (Supplementary Fig. 6B–D), this corresponded to an inoculum of ~$10^7$ bacteria and an MOI of 1:10. UPEC was then allowed to grow undisturbed for 3–10 h, depending on experimental conditions. For Supplementary Fig. 7E, the mini-bladder was lumenally pre-treated with 0.5 mM EDTA diluted in low solute SHU for 30 min before infection.

## Micturition in the mini-bladder and CFU measurements

To simulate micturition, bacteria that had grown in the lumen were first removed by pipetting out the contents from the outlet port and collecting the washout. Thereafter, 100 µL of sterile urine each was perfused through the lumen twice and the washout collected. The total washout was then plated for CFU. This is referred to as the CFU count pre-micturition. Thereafter, a further two washes with 100 µL of sterile urine each were performed, and the washout collected to enumerate any remaining UPEC in the lumen. This is referred to as the CFU count post-micturition. CFU was counted by plating on LB or LM[32] agar plates and incubating overnight at 37 °C. LM agar was made by diluting 120 g D (+)-sucrose (0.58 M), 22.2 g Bacto Brain heart infusion (Thermofisher; 237300), 6 g Bacto Agar (Becton Dickinson; BD0214010), 1.2 g $MgSO_4$ in 600 mL of $ddH_2O$. For experiments to test the impact of the D-mannose on UPEC adhesion, D-mannose was added in PHU at a final

concentration of 11.1 mM, and this urine formulation was used for infection and washes.

## Staining exfoliated epithelial cells

Exfoliated epithelial cells were collected by centrifuging (600 g, 10 min) the pre and post micturition washouts. The pellet was resuspended in 100 µL of PBS. This suspension was fixed with 4% paraformaldehyde (PFA; Thermofisher) in PBS overnight at 4 °C. The sample was washed (by centrifugation, 600 g, 10 min) with PBS, and permeabilized with 0.1% Triton X-100 (Sigma-Aldrich) in PBS (Gibco) (30 min, RT). This was followed by blocking in 1% Bovine Serum Albumin in PBS (Gibco) (blocking buffer, 1 h, RT). The samples were stained for DAPI (1: 300, Thermofisher, 62248) and Alexa Fluor 555–Phalloidin (1:400, Thermofisher, A34055) diluted in blocking buffer for 1 h at RT. The exfoliated epithelial cells were resuspended in 50 µL of PBS and placed on a slide for imaging.

## Staining proliferating and dead cells in the mini-bladder

For live-imaging dead cells (e.g., Fig. 2B), the tissues were exposed to DRAQ7 (1:1000, Thermofisher, D15106) diluted in urine and the basal media for 30 min at 37 °C pre-imaging. Overall nuclei were stained for live-imaging using Hoechst solution 33342 (1: 1000, Thermofisher, 62249) diluted in urine and the basal media for 1–2 h at 37 °C.

For staining proliferating cells (e.g., Supplementary Fig. 4D), the tissues were exposed to 10 µM of EdU labelling solution (Click-iT Plus EdU Imaging Kit, Thermofisher) diluted in urine and the basal media for 3 h at 37 °C. The tissues were then fixed with 4% paraformaldehyde (PFA; Thermofisher) in PBS overnight at 4 °C. The tissue was permeabilised with 0.1% Triton X-100 (Sigma-Aldrich) in PBS (Gibco) (30 min, RT). This was followed by blocking in 1% Bovine Serum Albumin in PBS (Gibco) (blocking buffer, 1 h, RT). To detect EdU, the Click-iT® Plus reaction cocktail was freshly prepared (as per the recommended protocol by the manufacturer; 0.5 mL cocktail was made with 440 µL 1X Click-iT® reaction buffer, 10 µL copper protectant, 1.2 µL Alexa Fluor® picolyl azide, 50 µL 1X Click-iT® EdU buffer additive). The tissue was exposed to 0.5 mL of Click-iT® Plus reaction cocktail for 30 min at RT.

## Antibiotic treatment in mini-bladder

For antibiotic treatment, Fosfomycin (300 µg/mL, 40X MIC in high solute SHU; 300 µg/mL, 20X MIC in PHU high) or Ciprofloxacin (5 µg/mL, 100X MIC in high solute SHU) diluted in sterile urine and basal media was introduced into the devices. The total UPEC remaining post treatment was quantified by collecting urinal bacteria with 2 washes of 100 µL sterile urine with antibiotics. The lumen was then replaced with sterile urine and the basal chamber with basal media, both without antibiotics to allow bacterial regrowth. For the experiment with Fosfomycin treatment in Fig. 4F, the lumen was washed twice with 100 µL of sterile urine without antibiotics at different timepoints during the regrowth phase, and the washout was used for counting CFU.

## Microscopy and image processing

**Immunostained mini bladder tissues.** Immunostained tissue was treated with FocusClear (CelExplorer, FC-101) for 30 min at RT to enhance the transparency of the hydrogel. The sample was then mounted on an ibidi dish with 90% Glycerol in PBS (Gibco).

For whole tissue imaging, related to Fig. 1B, C, E, G; Supplementary Fig. 2A, the sample was imaged using an inverted Leica SP8 confocal microscope with a white light laser and 25x water immersion objective (NA = 0.95, Leica). The voxel size was 0.91 µm × 0.91 µm × 1 µm in X, Y, and Z. The laser lines 405 nm, 488 nm, 555 nm, and 647 nm were used with intensity between 5 and 20%, adjusting for image saturation. The whole tissue has a thickness of ~300 µm in Z. Owing to scattering within the tissue, it was not possible to image the whole tissue with the same

illumination settings. Instead, the tissue was imaged in 3 sections, each of 100 μm thickness in the Z plane. To account for the increased scattering within the deeper sections of the tissue, the overall laser intensities were adjusted for each section to maintain similar image saturation. The 3 sections were then stitched together along the Z axis using ImageJ to obtain the overall shape of the tissue in 3D. Background subtraction of 50 pixels and a median filter of 0.5 pixels was applied to the stack in ImageJ. Imaris 9.9 (Bitplane) was used for rendering 3D images and orthogonal views.

For magnified sections, related to Supplementary Fig. 2B–H, Fig. 2G and Fig. 1D, the sample was imaged using an inverted Leica SP8 confocal microscope. Supplementary Fig. 2B was imaged white light laser and a 40x glycerol immersion objective (NA = 1.25, Leica). The pixel size was 0.14 μm × 0.14 μm in X-Y. The laser lines 405 nm and 555 nm were used with intensity between 5 and 20%, adjusting for image saturation. Supplementary Fig. 2C, H and Fig. 2G were imaged with a white light laser and 40x glycerol immersion objective (NA = 1.25, Leica). The voxel size was 0.28 μm x 0.28 μm x 0.36 μm in X, Y, and Z. The laser lines 405 nm, 488 nm, and 647 nm were used with intensity between 5 and 20%, adjusting for image saturation. Supplementary Fig. 2 D, E, F, G were imaged with a white light laser and 25x water immersion objective (NA = 0.95, Leica). The voxel size was 0.23 μm x 0.23 μm x 0.57 μm in X, Y, and Z. The laser lines 405 nm, 488 nm, 647 nm and 696 nm were used with intensity between 5 and 20%, adjusting for image saturation. Figure 1D was imaged white light laser and a 63x oil immersion objective (NA = 1.40, Leica). The pixel size was 0.3 μm × 0.3 μm in X-Y. The 405 nm and 555 nm laser lines were used with intensity between 5 and 20%, adjusting for image saturation. Imaris 9.9 (Bitplane) was used for rendering 3D images.

For infected tissue, related to Fig. 5C, D, E and Supplementary Fig. 6E, F, the sample was imaged using an inverted Leica SP8 confocal microscope. Supplementary Fig. 6E, F were imaged with a white light laser and 63× water immersion objective (NA = 1.20, Leica). The voxel size was 0.18 μm × 0.18 μm × 0.36 μm in X, Y, and Z. Figure 5C, D, E were imaged with a white light laser and 63x water immersion objective (NA = 1.20, Leica). The voxel size was 0.36 μm × 0.36 μm × 0.36 μm in X, Y, and Z plane. The 405 nm, 488 nm, and 555 nm laser lines were used with intensity between 5 and 20%, adjusting for image saturation. Imaris 9.9 (Bitplane) was used for rendering 3D images.

For EdU measurements, related to Supplementary Fig. 4D, the sample was imaged using an inverted Leica SP8 confocal microscope with a white light laser and 25× water immersion objective (NA = 0.95, Leica). The pixel size was 0.45 μm × 0.45 μm in X-Y with 1 μm in Z plane. The laser lines 405 nm and 647 nm were used with 10% intensity. Imaris 9.9 (Bitplane) was used for rendering 3D images (surface grain size 1 μm; auto thresholding) and analysing the volume of EdU+ and Hoechst+ sections.

**Live imaging DRAQ7+ cells in mini bladder tissues.** For live-imaging DRAQ7+ cells in the mini bladder, related to Fig. 3H and Supplementary Fig. 4B, the sample was imaged using a Leica Thunder widefield imaging system with a 25× water immersion objective (NA = 0.95, Leica). The voxel size was 0.52 μm × 0.52 μm × 1 μm in X, Y, and Z. The LED source was used with 5–10% and 16% intensity, respectively, with bandpass dichroic filters at 405 nm and 647 nm. The samples are imaged to capture a Z-stack of 100 μm from the bottom. The image was deconvoluted using Leica Thunder software with Large Volume Computational Clearing (LVCC). Imaris 9.9 (Bitplane) was used for rendering 3D images (surface grain size 1 μm; auto thresholding) and analysing the volume of DRAQ7+ and Hoechst+ sections.

**Dextran based diffusion assay.** The mini-bladder lumen was perfused with fluorescein isothiocyanate (FITC)-tagged dextran (average

mol wt. 4000, Sigma Aldrich, 46944) diluted to 1 mg/mL in PBS (related to Figs. 1H, 3F). The centre Z-slice of the tissue (approx. 150 μm from the bottom of the lumen) was imaged immediately with Leica Thunder widefield imaging system using a 25x water immersion objective (NA = 0.95, Leica). The pixel size was 0.26 μm × 0.26 μm in X-Y. The LED source was used at ca. 5–10% intensity adjusted for saturation with bandpass dichroic filters at 475 nm. Intensity variation was plotted at multiple cross-sectional lines perpendicular to the urinal axis. At most 3 cross-sectional lines were measured per device. The intensity is normalised between the maximum and minimum values. The normalised intensity values at $Z = 525$ μm away from the central urinal axis on both sides are plotted for the relative dextran permeability graph (in Figs. 1I, 2H, 3G).

**Time lapse of infected mini bladder tissue.** Timelapse snapshot images of infected tissue samples in Figs. 3B, 4B, Supplementary Fig. 6B, Supplementary Fig. 7B were imaged with Leica Thunder widefield imaging system using a 25x water immersion objective (NA = 0.95, Leica). The stage-top incubator was connected to a gas mixer (Okolab) to maintain 5% $CO_2$ and a temperature of 37 °C throughout the imaging period. For Fig. 3B, the voxel size was 0.26 μm × 0.26 μm × 1 μm in the X-Y and Z plane. The LED source was used with 10% intensity with bandpass dichroic filters at 475 nm. For Fig. 4B, Supplementary Fig. 6B (UTI89), Supplementary Fig. 7B, the pixel size was 0.52 μm × 0.52 μm in X-Y with 1 μm in Z plane. The LED source was used with 10–15 % intensity with bandpass dichroic filters at 475 nm, intensity adjusted for saturation.

Timelapse snapshot images of infected tissue samples in Fig. 4E, Supplementary Fig. 6B (CFT073) were imaged using an inverted Leica SP8 confocal microscope using a 25x water immersion objective (NA = 0.95, Leica). The stage-top incubator was connected to a gas mixer (Okolab) to maintain 5% $CO_2$ and a temperature of 37 °C throughout the imaging period. For Supplementary Fig. 6B (CFT073), the voxel size was 0.91 μm × 0.91 μm × 1 μm in X, Y, and Z. The laser line 500 nm was used with 6% intensity. For Fig. 4E, the voxel size was 0.45 μm × 0.45 μm × 1 μm in X, Y and Z plane. The laser line 500 nm was used with 15% intensity.

All samples were also imaged using brightfield illumination. A total Z-stack of 120 μm from the bottom was captured. Tissue-associated bacteria were analysed using Imaris 9.9 (Bitplane) (an example of the analysis pipeline is shown in Supplementary Fig. 10). Imaris 9.9 (Bitplane) was used to create a surface using the bacterial signal in a 650 μm × 650 μm × 100 μm cuboid centred on the bottom layer of tissue (surface grain size 0.5 μm; auto thresholding). The total volume of the surface was exported as the tissue- associated bacterial volume.

**Bacterial imaging.** Bacteria related Supplementary Fig. 6G, H, Supplementary Fig. 8L and Supplementary Fig. 9B were imaged using an inverted Leica SP8 confocal microscope. Supplementary Fig. 6G, H were imaged with a white light laser and 63x oil immersion objective (NA = 1.4, Leica). The pixel size was 0.1 μm × 0.1 μm in X and Y. Supplementary Fig. 8L was imaged with a white light laser and 63x oil immersion objective (NA = 1.4, Leica). The pixel size was 0.14 μm × 0.14 μm in X and Y. Supplementary Fig. 9B was imaged with a white light laser and 63x oil immersion objective (NA = 1.4, Leica). The pixel size was 0.36 μm × 0.36 μm in X and Y. The 488 nm laser line was used with intensity between 5 and 20%, adjusting for image saturation.

Bacteria related to Fig. 5B and Supplementary Fig. 8C were imaged using Leica Thunder widefield imaging system. Figure 5B was imaged with 40x dry objective (NA = 0.95, Leica). The pixel size was 0.32 μm × 0.32 μm in X and Y. Supplementary Fig. 8C was imaged with a 40x dry objective (NA = 0.95, Leica). The pixel size was 0.16 μm × 0.16 μm in X and Y. LED source was used at 5–20% intensity with bandpass dichroic filters at 475 nm.

## Array tomography SEM: sample preparation and imaging (stretched and unstretched tissue)

For array tomography SEM of stretched and unstretched tissue (Fig. 1J, Supplementary Fig. 3), a fixative solution of 1% PFA and 2.5% glutaraldehyde in 0.1 M phosphate buffer was added to the basal side reservoirs while tissues were stretched to their maximum state or in unstretched conditions. The tissues were then left for at least 2 h in the "inflated" or normal state while connected to the syringes. Samples were then incubated for 1 h in 2% (w/v) osmium tetroxide and 1.5% (w/v) $K_4[Fe(CN)_6]$ in 100 mM PB buffer. Samples were incubated for 1 h in 1% (w/v) tannic acid in 100 mM PB buffer, then 30 min in 2% (w/v) aqueous solution of osmium tetroxide, followed by 1% (w/v) uranyl acetate for 2 h at room temperature. At the end of gradual dehydration cycles, samples were flat-embedded in Epon-Araldite mix.

Polymerised flat blocks were trimmed using a 90° diamond trim tool (Diatome, Biel, Switzerland), and the arrays of 100 nm sections were obtained using a 35° ATC diamond knife (Diatome, Biel, Switzerland) mounted on a Leica UC7 microtome. Arrays were transferred on wafers as described in the following protocols[79,80].

Wafers were analysed using Helios SEM microscope (Thermo Fisher Scientific) at 2 keV landing energy and 0.8 nA beam current at 2 mm distance using a Mirror Detector (MD-BSA). Images were collected semi-automatically with 5 µs dwell time using Maps 3.11 software (Thermo Fisher Scientific). Images were collected at or 5 mm/s ($6084 \times 2044$) to generate 5 nm resolution images, respectively. To cover a larger area of the sample, several images were collected at a given resolution and stitched together using Maps 3.17 software (Thermo Fisher Scientific)[79,80].

**Analysis of vesicles.** Vesicles in array tomography SEM cross sections were identified manually based on contrast differences (example—Supplementary Fig. 3C, D). The total number of vesicles within each lumen facing the cell was enumerated manually in both stretched and unstretched tissue (Fig. 1K).

## TEM: sample preparation and imaging (axenic CWD UPEC)

For transmission electron microscopy of axenic UPEC samples (Supplementary Fig. 8I, J), the bacteria pellet was resuspended in a buffered mix of 1.0% glutaraldehyde and 2.0% paraformaldehyde in 0.1 M phosphate buffer, pH 7.4 for 2 h. After gentle centrifugation, the pellet was then postfixed 1.0% osmium tetroxide with 1.5% potassium ferrocyanide, and then 1.0% osmium tetroxide alone. It was finally stained for 30 min in 1% uranyl acetate in water before being dehydrated through increasing concentrations of alcohol and then embedded in Durcupan ACM (Fluka, Switzerland) resin before being placed between glass slides, and the resin was cured at 60 °C for 24 h. Once hardened, the resin-embedded pellet was glued to blank resin blocks with cyanocrylate glue, and thin (50 nm thick) sections were cut with a diamond knife. These were collected onto pioloform support films on single slot copper grids, contrasted with lead citrate and uranyl acetate. Sections were imaged in a transmission electron microscope (FEI Tecnai Spirit) operating at 80 kV using a digital camera (FEI Eagle).

## Serial block face-scanning electron microscopy (SBEM): sample preparation of mini-bladder tissue

Mini-bladders at 22hpi post Fosfomycin treatment (Fig. 5F–J) were fixed in 1% glutaraldehyde and 2% paraformaldehyde in 0.1 M phosphate buffer at pH 7.4 for 1 h. The hydrogel block containing mini bladder tissue was extracted by cutting the PDMS using a razor blade. A diagonal cut was made across the sample to ensure complete flow of fixative to the apical side of the tissue. The sample was then blocked in 1% Bovine Serum Albumin in PBS (Gibco) for 30 min followed by staining with Hoechst solution 33342 (1: 300, Thermofisher, 62249) and Alexa Fluor 555—Phalloidin (1:400, Thermofisher, A34055) for 1 h at RT. Post wash, the sample was screened using Leica SP8 confocal microscope with a 63× water immersion objective to identify suitable sites of CWD bacteria colocalising with the urothelial cells and image series were collected through the entire structure. The sample was left overnight in the same fixative and then post-fixed in potassium ferrocyanide (1.5%) and osmium (2%), followed by thiocarbohydrazide (1%), and then osmium tetroxide (2%). It was then stained overnight in uranyl acetate (1%), washed in distilled water at 50 °C, before final staining with lead aspartate at the same temperature. They were finally dehydrated in increasing concentrations of ethanol and then embedded in Durupan resin and hardened at 65 °C for 24 h between glass slides.

## Serial block face electron microscopy (SBEM)

The resin-embedded device was glued with conductive cement to an aluminium stub oriented in the same way as when it was imaged in the confocal microscope. It was then trimmed to expose only the region of interest for imaging in the electron microscope. This region was approximately $300 \mu m \times 300 \mu m$ in the $x$ and $y$ directions. Care was taken to compare images of the block with those taken in the confocal microscopy so that the corresponding regions could be imaged. The final block was mounted inside a scanning electron microscope (Zeiss Merlin, Zeiss NTS), holding a block face cutting microtome (3View, Gatan). Layers of resin, 50 nm thick, were cut from the block surface, and sequential images collected after each layer was removed. An acceleration voltage of the 1.6 kV was used with a pixel size of 8 nm with a dwell time of 1 µs. Series of nearly aligned images was collected, with final alignment being carried out in the FIJI imaging software (www.fiji.sc). Segmentation of different cells and structures was carried out on the Webknossos platform. Final models were exported to 3D modelling software for reconstruction (Blender.org).

## Quantification and statistical analysis

All statistical analysis was performed using GraphPad Prism. Details of the statistical tests and exact numbers of biological and technical replicates are provided in the figure legends, which also provide information about the depiction of the mean or median values and standard deviations or standard errors of the mean and confidence intervals where appropriate.

## Reporting summary

Further information on research design is available in the Nature Portfolio Reporting Summary linked to this article.

## Data availability

The data generated in this study are provided in the Supplementary Information/Source Data file. The RNAseq raw data uploaded to Zenodo can be accessed via https://doi.org/10.5281/zenodo.14802031. Source data are provided with this paper.

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

## Acknowledgements

The authors wish to thank J. Sordet, B. Mangeat and members of the Histology, Gene Expression and BioImaging and Optics Platform Core Facilities at EPFL; T. Simonet, F. Normandeau for experimental assistance and the lab of Prof Calin Guet (IST, Austria) for the lab strain UTI89. K.T. acknowledges support from a Swiss Government Excellence Scholarship (2022.0289), a Swiss National Science Foundation (SNSF) Postdoctoral Fellowship (TMPFP3_217144) and support from the IBSA Foundation. This research was supported by grants to J.D.M. from the National Centre of Competence in Research (NCCR) AntiResist funded by the SNSF (grant number 51NF40_180541), and from the Fondation Leenaards. M.N., N.G., and M.P.L. acknowledge support from EPFL. V.V.T. acknowledges support from the Novartis Foundation for Medical-Biological Research (#20C240), the Holcim Stiftung zur Förderung der Wissenschaftlichen, the Heidelberg University Medical Faculty and a Life Sciences Bridge Award from the Aventis Foundation. Schematics were created using BioRender.com (McKinney, J. (2026) https://BioRender.com/55huf39).

## Author contributions

Conceptualisation: G.P., M.N., K.S., and V.V.T.; methodology: G.P., M.N., K.S., N.G., and V.V.T.; formal analysis: G.P., J.B., G.A., I.K., G.W.K., and V.V.T.; investigation: G.P., M.N., K.S., J.B., K.T., L.I.E.S., V.B., G.A., I.K., and S.C.R.; resources: V.V.T., M.L., J.D.M.; writing—original draft: G.P. and V.V.T.; writing—review and editing: G.P., M.N., K.S., K.T., V.B., G.A., I.K., G.W.K., M.L., and V.V.T.; visualisation: G.P., J.B., G.A., I.K., G.W.K., and V.V.T.; supervision: M.L., V.V.T., and J.D.M.; project administration: V.V.T. and J.D.M.; funding acquisition: M.L., V.V.T., and J.D.M.

## Competing interests

M.N., N.G., and M.L. are current employees of Hoffmann-La Roche Ltd or were employed by the company while working on this study. The company provided support in the form of salaries for authors but did not have any additional role in the study design, data collection and analysis, decision to publish or preparation of the manuscript. The remaining authors declare no competing interests.
