## [Transparent Peer Review file · Nature Communications]

A microphysiological human mini-bladder reveals urine-urothelium interplay in tissue resilience and UPEC recurrence in urinary tract infections

Corresponding Author: Dr Vivek Thacker

Version 0:

Reviewer comments:

Reviewer #1

(Remarks to the Author)

In this study, Paduthol & Nikolaev et al. developed a mini-bladder model that simulates urine-urothelium interactions, incorporating a stratified urothelium and micturition. Using this system, the authors demonstrate that exposure to high-solute urine disrupts tight junctions, alters immune responses, and reduces tissue resilience, increasing susceptibility to uropathogenic *Escherichia coli* (UPEC) colonization. They also show that bacteria evade killing by ciprofloxacin and fosfomycin in this environment and that fosfomycin promotes the formation of cell wall-deficient UPEC, which could contribute to recurrent infections. This model is novel and represents a significant advancement in the field. The results are compelling and of interest to the field.

Major comments:

-The term organoid refers to self-organized three-dimensional tissue cultures that are derived from stem cells. Normally these are grown in 3D dimensions (in ECM – e.g. Matrigel). I find it misleading since the authors use here a primary cell line.

-Line 175 “the mini-bladders exposed to low solute SHU continued to support active proliferation”: The images show that the mini-bladders are not destroyed by the low solute SHU but they do not show active proliferation (e.g., EdU staining). In fact, if the model accurately mimics the bladder, proliferation should be very low after differentiation and in the absence of any insult.

-Line 244: “which can be explained by improved access to deeper layers of the tissue caused by the loss of tight junctions...”. The authors do not demonstrate whether the loss of tight junctions is directly related. They could either provide evidence (e.g., loosening tight junctions with EDTA in the presence of low or mild solute) or adjust the wording to indicate that it “could” be explained by this mechanism.

-In Fig. 4: the authors state that when infecting with high or low solute SHU, 6 hpi is already sufficient to observe differences in tissue-associated bacteria. Do tight junctions disrupt that quickly? Only nuclei counts are shown in Fig. 2 (for SHU) and Fig. S2 (for PHU). Given the importance the authors place on tight junction integrity throughout the manuscript, it would be helpful to include ZO-1 staining over time to better assess these changes.

-It is unusual that the mini-bladders were grown in different media depending on their appearance. The authors mentioned that in about 10% of the chips, they had to supplement the medium with growth factors (Y-27632, A83-01, CHIR99021, B27, FGF7, 10, and 2). This could have significant effects on the tissue or its response, as for example, Y-27632 and A83-01 are inhibitors of key host pathways. It may also explain why some replicates appear very different from others in some experiments. Defective mini-bladders should be excluded from the analysis.

Minor comments:

- Fig.1. (C,E,G):A close-up in the figures would improve visualization.
- Since the quantification is image-based for tissue-associated bacteria, it would be helpful to include some close-up images. Were these quantified from images similar to those in Fig. 3, 4, S5, and S6?
- It would be helpful to see the DRAQ7 staining in the mini-bladders exposed to the PHU (Fig. 2S) to compare it with those in Fig. 2 with SHU.
- I am sure there is a reason for using SHU or PHU in different experiments, but it is not always clearly stated in the text. Some clarification would help the reader understand.
- Does the permeability of the model change with exposure to high-solute PHU or SHU?
- It is interesting that while the bacterial doubling time in SHU remains the same between high and low solute conditions, in PHU, there is a significant difference between high and low solute but not as much between mild and high. Could this be due to a lack of nutrients in the low (more diluted) solute PHU?
- Representative images corresponding to the measurements in Fig. S6A-B would be good as well.
- Reference 46 is missing.

Reviewer #2

(Remarks to the Author)

The paper from Paduthol and colleagues presents an intriguing new human urothelial model, which is used to explore several interesting biological questions including the effect of urine concentration on tissue damage and antibiotic response. It's a tremendous achievement to manage to get (mostly) differentiated, urine-tolerant urothelium, flow and stretch all into one model, which is a great advance from this lab's previous work with Emulate. Overall, the model has potential for discovering some worthwhile new host/pathogen biology beyond what has been explored here, and is impactful for the field. There are, however, some concerns, both major and minor, which we've outlined below in the hopes that these comments might prove useful for improving the manuscript.

Major concerns

1. The premise of the work is based on literature which the authors use to lay out as a general fact that high-solute urine damages the urothelium ("gradual tissue death", tight junction depletion, apoptosis etc.). I was surprised to see this as I was not familiar with such facts, especially as it seems counterintuitive that any sort of urine would gradually kill the very organ that has been highly evolved and equipped to deal with it. Dehydration would have been a common occurrence in the human race (and all mammals) over millennia, and still is. I looked up each of the papers supporting the idea that urine degrades the bladder (13,14,15,16 and 43). Three of the studies were in rats, one was in rat cell culture, and the last was in toads. Cell cultures cannot tolerate urine, not having the requisite protection, so this may not be the best model system to study cytopathic effects of urine. I'm not familiar with the toad urothelium, but it's well known that rodents have more concentrated urine than humans, and of course they have thinner urothelia, which might make them more delicate. I think it's misleading to lay out these "facts" without acknowledging that they are done in model systems that might not be as relevant to the human bladder – the way it is phrased, the reader may assume these are human studies. I certainly did before checking the references.

2. Line 117. "The resultant tissue at day 20 recreates the organization of the human bladder epithelium." Authors are entirely silent, both here in the Results and later in the Discussion, on the fact that the normal urothelium does not have stretches of undifferentiated, CK20-negative tissue punctuated by blooms of differentiated (possibly hypertrophic?) tissue. The human urothelium should have a uniform layer of intermediate cells 5-6 layers deep. In the example shown in Fig 1, it looks like this ranges from 0 to 5 intermediate cells depending on location. Don't get me wrong; this model is still an impressive achievement, but its limitations need to be acknowledged up front in the results text and not waved away under the guise of a perfect "recreation". The fact that a significant subset of the bladder urothelia is undifferentiated also makes us take the high-solute damage with a grain of salt – no pun intended. Maybe these undifferentiated zones are more prone to infection as well.

Given the above, please qualify all sentences such as that in line 151-2: "Together, these results confirm that the mini-bladder microtissue model contains a stratified, differentiated urothelium [with the implication that this is 100% true across the tissue]..." See also Discussion, line 344.

3. Line 162. Experiments are said to be done here with mini-bladders that had "differentiated uniformly": this is interesting, as it confirms that perhaps they often don't, as demonstrated in Figure 1 – but we should be shown an image or two of fully/uniformly differentiated and stratified bladders in Figure 1 as well, just so we can see for ourselves. By extension, are we to assume that all the other experiments were done with the incompletely developed mini-bladders shown in Figure 1?

4. Fig 1K. Knowing whether fusiform vesicles are present in the model, meaning that the recycling that has been shown to allow bacterial entry during the stretch/relax cycle (e.g. Truschel et al.) might take place, is crucial to understand how physiological your dynamic bladder model is. We as readers cannot be satisfied with Figure 1K without seeing any images of these vesicles. There is nothing evident in Figure 1J – not surprising at that resolution. Representative images of the correct resolution that were used to quantify these vesicles *absolutely must be shown* - in Supplementary is fine. At such resolution, it would also be possible to see hinges/asymmetric unit plaques by EM and this would also be very useful to confirm physiological relevance – as you are showing evidence of damage from high-solute urine, one could argue that this isn't surprising if the model lacks AUM plaques. The authors should also confirm the presence of the uroplakins by IFA – it is

a straightforward thing to check as there are a number of great antibodies. As the authors will know, the uroplakin complex will not form properly unless all components are present and correct – only UP3 RNA was tested. The question of whether intact uroplakin complexes are elaborated on the surface and make AUM is arguably more important for infection studies and differentiation status than is CK20. If you tried to see UP3 or other UPs by IFA and the model does not express them on the surface, this needs to be admitted. We aren't shown any higher resolution top-down confocals with actin either – do you see the expected enlarged, honeycomb-like pattern? Bottom line, are these apical cells truly terminally differentiated umbrella cells or not?

5. Regarding the solute status in human and synthetic urine – matching osmolality seemed to be your only focus. But as I understand it, the kidneys concentrate different solutes differently; for example, urea is subject to disproportionate concentration compared with other chemicals. Just diluting globally to the right osmolality implies that it is only osmolality per se that governs any downstream effects, when actually it might be the concentration of one or a few key chemicals, which again are not proportionally concentrated by dehydration. I think getting the proportions right is crucial – which could be found in the literature, or by analyzing the components of the human urine you are using, and making informed choices how to make up the synthetic high and low urines. If my statements are incorrect, perhaps insert a line in the paper about how your choices were made.

6. In a related vein, pH is known to affect bacterial behavior as well as drug action. Why have you made your synthetic urines so acidic compared with the urine you've collected from volunteers?

7. Limitations. In the discussion, it should be acknowledged that, from what is shown, the mini-bladder might not be uniformly stratified and differentiated in some/many experiments (depending on how often they look like Figure 1), and this state could absolutely affect how easily the tissue could be damaged by high solutes, as well as susceptibility to infection. It should be clear to the reader that you understand the limits of the model, and where there is room for future improvement.

Minor points

1. What sex are H-6215 primary human bladder epithelial cells? This should be specified in the Methods.
2. Line 687. "In case of any tissue defects or failure of stratification, post day 10 the basal media was supplemented with propagation factors..." This is a massive protocol difference between cultures with and without this propagation fudge factor. Experiments with one or other should be noted in the figures as the two types of tissue may not be comparable in all aspects.
3. What is the rationale for using high solute urine for differentiation? Sorry if I missed it.
4. MOI describes the probability of initial collision with surface host cells (Poisson distribution). Better to calculate MOI based on the estimated superficial cell number across the accessible surface area. Or at least give both figures so the reader can compare it to conditions in their own infection models.
5. Fig 1A. Add some direction of flow arrows into schematic just to make schematic clearer.
6. Fig 1G. ZO-1 is located at cell junctions. Hard to know if ZO-1 is expressed in between cells at this resolution. Could the authors use a higher objective for greater cell resolution and if it is logistically possible, image from top-down?
7. Fig. 1H & I. Is this from stretched or unstretched samples? And would this change with the physiological difference?
8. Fig 1F. The assay used to quantify "expression" is not mentioned in results text or figure legends. Readers should not have to trawl through the methods to find out if this is protein level (image quantification) or RNA. I couldn't find it in the methods either. As there is some RNA work in the paper, and there are no IF images of the parental cell line, perhaps it's RNA expression? Entirely unclear. I may have just missed it, but figure legends should tell you what sort of assay it is at the bare minimum. "RE units" on the Y axis could be any sort of arbitrary unit including fluorescence intensity so they don't shed light.
9. 131. Nit-picky point perhaps, but strong ZO-1 staining does not in fact confirm that the tissue retains a 'robust barrier'. Only the subsequent functional assay can.
10. Why didn't the authors do TEER? If the model is accessible to the probe, that would have been interesting, to compare with data from other models.
11. Fig 1J. The unstretched bladder EM looks very reminiscent of bladder biopsies. But in the stretched version, I'm not really seeing the flattened umbrella cells I'd expect. The image is very limited (2-3 superficial cells). Could further examples be presented (even if in Supplementary) so we can view the 'average' superficial cell morphology? This goes back to major point 4, above.
12. All your tables are supplementary but in the text they are referred to as Table 1, Table 2 etc., not S1, S2, which sent me on a fruitless hunt.

13. Figure 2. Given that an infected urothelium will shed cells, can your bulk transcriptomics distinguish between “down-regulation” of proteins vs loss through absence of cells? All your effects are described as downregulation. Line 181-2 – ‘Long-term exposure to high solute SHU led to lower expression of cell-cell adhesion and proliferation gene sets (Fig 2E)’. Sorry if I’m missing something, but if high solute PHU is important for proliferation of immature bladders then how can SHU high /low solute then lead to lower/higher expression cell adhesion and proliferation, respectively?
14. Line 201-202 – Is there difference to CFT073 if pre-exposed to low and high solute urine?
15. Line 208 – IBC formation. Could Figure S4E show an orthogonal view to show depth of IBCs?
16. Figure 3B – Please confirm laser power for confocal microscopy is the same for all images?
17. Line 215. “but washing did not completely clear the lumen. This is not unexpected as bacteria near the epithelial layer boundary experience lower urine flow rates...” The context around this point could be more nuanced given that flow has actually been shown to increase UPEC adhesion, which isn’t mentioned here.
18. The word ‘axenic’ is not perhaps the best choice choice, as it is usually used to reference single-species growth in comparison to polymicrobial growth with other bacteria or parasites, which is not studied in this paper. I took a survey on my floor and only about 2 people out of 20 had even ever seen this word before. Most people in the field would say broth culture?
19. Line 289. Technically speaking, a still image cannot distinguish whether a bacterium is going from coccus to bacillus or vice versa.
20. Line 292. Here or in the discussion, you should explore the relationship between the CWD spherical E. coli you saw, versus the coccus-shaped E. coli found in mouse IBCs. Do you think these are totally different beasts?
21. Line 353. Seems to imply that hydrated individuals would not get UTIs. That is likely not true: rephrase. Also begs the question why rodents, with their more concentrated urines, are difficult to give a UTI. Perhaps this should be discussed.
22. Is there a difference between stretched and unstretched bladder with regard to infection? Maybe more clarity needed in text whether differences are seen with this phenotype.
23. Figure 4 - Is 2.5 hours enough to show regrowth with ciprofloxacin? For Fosfomycin, in low solute urine would regrowth rebound to higher CFU levels if more time for regrowth were allowed?
24. Line 281 – Will D-sucrose or BHI affect growth by introduction of new carbon sources, which may explain higher urinal CFUs in figure S6C & D?
25. Would you find higher CFU counts 29 hpi in tissue associated CFUs as CWD may have invaded into bladder? Only urinal UPEC tested (Figure S6 E & F).
26. Under what ethical framework were patients consented for urine? Please give IRB/ethics committee and license number in the Methods as is standard, or insert a statement why this is not necessary (perhaps they do things differently in Switzerland than in the rest of the world?)

Reviewer #3

(Remarks to the Author)

Reviewer #4

(Remarks to the Author)

A bioengineered human urothelial organoid model reveals the urine-urothelium interplay in tissue resilience and UPEC recurrence in urinary tract infections

In this study Paduthol et al. have developed a urothelial organoid model to study tissue resilience and bacterial recurrence during model UTI.

Building on previous work from this group, and others, in this study the authors have combined various important aspects of the infection cycle from a host perspective – e.g. stratification and dynamics of micturition – in a single system.

Overall, I find this work to be of technically high standards with complementary approaches used, the biological leaps are perhaps a bit less advanced but would likely be interesting for an audience relevant to Nature Communications.

Apart from a few general issues (outlined below), I think this study is a significant step forward in the infection-on-a-chip model development and could be accepted, pending satisfactory answers to the comments.

General/model comments:

- The figures are quite busy, and figure legends not always clear to follow. Could this be looked over throughout.
- The superficial layer(s?) of cells seem quite uneven (Figure 1C,E,G), how did that correlate with bacterial invasion? Did the authors notice if the green blobs (multiple IBCs? Figure 3B) colocalised with these protrusions of bladder cells?
- One possibly important issue is that the UPEC strain CFT073 is used, this strain could be viewed as substandard when it comes to study cystitis (common strain would be UTI89), as CFT073 was isolated from a pyelonephritis patient. I believe this study would be significantly stronger if data from an UTI89 strain was to be included. The experiments should at the very least be repeated with a Δ fimH (main adhesion protein in bladder infection) or Δ papG (main adhesion protein in kidney infection) strain, i.e. CFT073 Δ papG, to show that they system behaves like previously shown *in-vivo* systems.
- Line 178: RE: transcriptional profiling of mini-bladder tissue exposed to high- and low-solute SHU. A comparison should be made to the qRT-PCR expression profiles from Fig. 1F. Were relative gene expression comparable for specific targets (e.g., P63 and others)? How would expression profiles differ (especially those related to chemokines and TLRs) between high-solute PHU when compared to high-solute SHU?
- Line 211: Can the authors provide more explanation into the micturition cycling events of the mini-bladder model? Previous models have utilised a urine-flow system which likely also mimics the micturition cycles within UTI, albeit less natural due to the constant flow.
- The amount of bacteria seems high (5×10^8 cell/mL, times 3 perfusions), can the authors comment on why they used this high numbers. To me this do not seem like a physiological bacterial load.
- There is a clear lack of immune cells in the current model. Addition of immune cells to this model would significantly strengthen the study. But I understand that doing so may require a fair amount of extra experiments. Maybe a consideration for the future, but the very least this should be expanded on in the discussion.

PHU vs SHU:

- For flushed bacteria, did the authors find filaments in the SHU experiments? And did they see differences between PHU and SHU (frequency, length, amount ect?), and between high and low solutes? If they saw no filamentation in the SHU, it would be a red flag of the chemical composition as it has been shown in various models (both in-vivo, in-vitro and human) that UPEC undergo morphology changes into a filamentous form during UTI when exposed to real human urine. I'd say it is essential to show that morphology change also in SHU.
- Another critical aspect of this is the growth of UPEC in PHU vs SHU. Others have in the past tried to formulate a SHU that is similar to that of a PHU, with varying success. Notably, if I recall correctly, I don't believe that Ipe et al, did that comparison in their paper (cited by the authors as source for their SHU). It needs to be shown that bacteria grow at a similar rate in the two, otherwise the comparison is not particularly informative. In addition, the pHs are different between the PHU and SHU (table S1), this too has been shown to affect UPEC growth and morphology in the past, but is not commented on (only how varying concentrations affect the bladder cells themselves, Lines 167-176).
- The initial bacterial burden is higher for high solute SHU, so this is perhaps not very surprising that the final bacterial burden is also higher, by the authors interpreted as lower antibiotic clearance (Figure 4D). Would the conclusions change if you for example normalised these data in order to compare?
- Line 202: Can the overall density (ρ) of the urine be determined and reported in Table 1? Previous studies have shown that the specific ρ of PHU impacts bacterial cellular fitness and survival within the intracellular bladder environment and may also impact the bladder ecosystem.
- Line 240: Can the authors give reasoning as to why they have used SHU instead of PHU for the antibiotic mini-bladder model assays? I understand that SHU is more robustly made and consistently produced for both high- and low-solute concentrations, but are additional crosstalks between cipro/fosfo treatment and the PHU that may have been missed?

CWD:

- Line 290: Given the fact that CWD UPEC can revert back to normal rod-shaped after antibiotic pressure is removed may suggest changes in gene regulation drive the CWD phenotype in the presence of fosfo. Can the authors consult the literature for transcriptomics studies with acute fosfo treatment in *E. coli* to better understand this mechanism or alternatively perform mechanistic analyses in their CFT073 model +/- fosfo treatment or in axenic high-solute culture. This will also give more functional insights into the persistence of CWD UPEC within the bladder tissue.

• Line 318: What is the proportion of CWD and non-CWD UPEC in high-solute SHU given the heterogenous nature of CWD bacteria?

Methods:

• Overall the model and biological methods sections are mostly well explained. There are issues with the imaging sections: they should be more detailed.

In general: not adequate description of fluorescence imaging, for example:

“The sample was imaged using Leica SP8 confocal microscope with a white light laser and 25x water immersion objective (NA = 0.95, Leica).” (Lines 803-804).

“Fixed samples (e.g., Fig 1D, 5E, F, S4E) were imaged using Leica SP8 confocal microscope with a 63x water immersion objective (NA =1.20, Leica)” (Lines 828-830).

What wavelengths were used, at what laser powers, pixel size, acquisition times, etc?

It would be impossible to repeat these experiment based on the information given currently.

• The images in Figures 3B, 4B and 4E are of varying quality. Particularly the fluorescence channel, it is hard to truly assess the data based on these images, was the fluorescence in the full volume of the images superimposed? I'm not questioning that the signal increased over time, looking at the images it does look like that qualitatively. But it is not immediately clear how quantification was done when comparing end point fluorescence.

There is no description of how they were acquired nor analyzed (or I couldn't find it at least. Lines 823-824 is not a well enough description of the analysis). Images like Figure 1D, including bacteria, clearly showing that they are intracellular show be included. Otherwise, how can the authors claim that the schematics only based on this model? This is also an excellent opportunity to show “how deep” the UPEC travels in the stratified layers.

Version 1:

Reviewer comments:

Reviewer #1

(Remarks to the Author)

The authors have made a significant effort to address the comments, and the manuscript has improved substantially. Most of my comments have been addressed appropriately, although I still have concerns with the following points:

- Related to Major Comment 1: I continue to disagree with the authors. The fact that other studies have used the term organoid incorrectly does not justify its use here. While organoids can indeed be derived from primary cells, these organoids are grown in extracellular matrix in a three dimensional context, which was not done here. Instead, the authors describe culture conditions that rely on standard cell culture medium and passaging with TrypLE Express. This is consistent with conventional primary cell culture rather than organoid culture. Therefore, the terminology should be corrected, and the term organoid should be removed from the title.
- Related to Major Comment 2: Supplementary Figure 4D does not show a representative image of the control (D0), which is the condition shown in the corresponding graph (nice data btw).
- Related to Major Comments 3 and 4: ZO1 staining is expected to localize at cell-to-cell junctions, not across the entire apical surface. In the new figure panels (Figure 2G and Supplementary Figure 2H), the ZO1 signal appears to extend across the whole apical membrane. Image resolution does not explain this observation, since the resolution should be comparable to that of the phalloidin images (Supplementary Figure 2B). The authors should clarify how they interpret this staining pattern.
- Additional Comment: In the new Supplementary Figure 2A and E, mature umbrella cells would typically appear as large polygonal cells positive for both UP3A and UP1A. In the new images, UP1A does not appear restricted to the superficial layer, and the expected large umbrella cells are not evident in panel D. Although some polygonal cells are visible in panel B, they do not seem large enough to represent fully mature umbrella cells. This is acceptable as no model is perfect, but it should be acknowledged as a limitation that the umbrella cell layer does not appear fully differentiated.

Reviewer #2

(Remarks to the Author)

The authors have clearly done a lot of additional work to make the manuscript more thorough and to make the conclusions more robust - or alternatively have caveated the language where appropriate.

Reviewer #3

(Remarks to the Author)

Reviewer #4

(Remarks to the Author)

All my points have been beautifully addressed and answered. I can only congratulate the authors on a great and exciting study.

Reviewer #5

(Remarks to the Author)

Point-by-point reply

We thank the reviewers for the constructive feedback to improve our manuscript. Guided by this feedback we have performed several new experiments and include additional data, as summarised below. In the manuscript file, the corresponding data and textual changes in the manuscript file are also highlighted in blue.

- 1) **Figure 1**: we have edited **panel A** to show direction of flow (requested by Reviewer 2/3)
- 2) **Figure 2**: **new panels G, H** - Immunostaining shows lower expression of ZO-1 in mini-bladders exposed to high-solute urine which is corroborated functionally with increased leakage using the dextran diffusion assay, demonstrating the loss of tight junctions (requested by Reviewer 1).
- 3) **Figure 3**: **new panels E, G** - New infection data with CFT073 Δ *fimH* strain shows that it phenocopies the exogenous addition of D-mannose with reduced tissue-associated bacteria, urinal bacterial burden, and tissue permeability (requested by Reviewer 4).
- 4) **Supplementary Figure 2**: High resolution images of the urothelial architecture and tissue differentiation (requested by Reviewer 1 and 2/3)
- 5) **Supplementary Figure 3**: Additional magnified array tomography SEM images of unstretched and stretched tissue cross-sections to visualise the tight junctions, vesicles and luminal cell morphology in stretched tissue (requested by reviewer 2/3).
- 6) **Supplementary Figure 4** (previously **Supplementary Figure 2**): **panel B, C** - Representative images for DRAQ7 staining in PHU-exposed mini-bladders confirm that outcomes with PHU are along the same lines as SHU. **Panel D, E** show the mini-bladders exposed to low solute SHU have a higher proportion of EdU+ proliferating cells (requested by reviewer 1).
- 7) **Supplementary Figure 5** (previously **Supplementary Figure 3**): **panel C** - A comparison of qRT-PCR expression profiles in the mini-bladder at D=0 with that in mini-bladders exposed to low and high solute urine for 6 days show comparable expression of targets (requested by Reviewer 4).
- 8) **Supplementary Figure 6** (previously **Supplementary Figure 4**): **Panel B-F** shows data from a 10-hour infection in the mini-bladder with UTI89 strain with PHU. **Panel G** shows filamentous UPEC collected in the effluent of mini bladders infected in high and low solute SHU (requested by Reviewer 4).
Panel E shows the cross section of an IBC (requested by Reviewer 2/3).
- 9) **Supplementary Figure 7** (previously **Supplementary Figure 5**): **Panel E** - mini-bladders treated with EDTA to weaken tight junctions show an increased burden of tissue-associated bacteria when infected in low solute SHU (requested by Reviewer 1).
Panel F - Normalised tissue associated bacterial volume related to **Fig 4D** shows no difference in the clearance rate or recovery of tissue-associated UPEC after antibiotic treatment, reinforcing the higher tissue-associated bacterial load accumulated prior to ciprofloxacin treatment as a driver for the differences between high and low solute SHU infection dynamics (requested by Reviewer 4).
- 10) **Supplementary Figure 8** (previously **Supplementary Fig 6**): **panel C, D** - representative images corresponding to the data in **panel A, B** (requested by Reviewer 1).
Panel L shows representative images of CWD UPEC generated with Fosfomycin in axenic culture in high solute PHU (requested by Reviewer 4).

- 11) Supplementary Figure 9 (previously Supplementary Fig 7): panel C - Quantification of the differential colony counts on LB and LM of axenic UPEC from high solute and low solute SHU shows the predominance of CWD UPEC in high solute SHU (requested by Reviewer 4).
- 12) Supplementary Figure 10: An illustrated example of the image processing pipeline used to quantify tissue -associated bacteria (requested by Reviewer 1 and 4).
- 13) Figure 3, D, F, G, Supplementary Figure 4 B, C, Supplementary Figure 7 B, C, D, Supplementary Figure 9 panel D, E: subset of data points corresponding to these figures were replicated and replaced to remove data from experiments where a different set of differentiation protocol was used (requested by Reviewer 1 and 2/3). New data agree with the previous versions of these panels in the original submission.
- 14) Materials and Methods: this section has been comprehensively edited to improve clarity and provide additional details where required to assist the reader in evaluating the experimental procedures described (requested by reviewer 2/3 and 4).

#Reviewer 1 (Remarks to the Author)

In this study, Paduthol & Nikolaev et al. developed a mini-bladder model that simulates urine-urothelium interactions, incorporating a stratified urothelium and micturition. Using this system, the authors demonstrate that exposure to high-solute urine disrupts tight junctions, alters immune responses, and reduces tissue resilience, increasing susceptibility to uropathogenic Escherichia coli (UPEC) colonization. They also show that bacteria evade killing by ciprofloxacin and fosfomycin in this environment and that fosfomycin promotes the formation of cell wall-deficient UPEC, which could contribute to recurrent infections. This model is novel and represents a significant advancement in the field. The results are compelling and of interest to the field.

AU response: We thank the Reviewer for this positive assessment of the manuscript and constructive suggestions which have improved the manuscript in revision.

Major comments:

1. The term organoid refers to self-organized three-dimensional tissue cultures that are derived from stem cells. Normally these are grown in 3D dimensions (in ECM – e.g. Matrigel). I find it misleading since the authors use here a primary cell line.

AU response: We appreciate the Reviewer’s point of view that describes perfectly the canonical picture of organoids. However, we do note that there are several instances of organoid cultures from primary cells, just like the ones we have used in this study. Furthermore, although the model does incorporate a defined 3D mould, the seeded cells differentiate, stratify, and form a three-dimensional tissue that looks completely different from the starting cells. We therefore think that the term organoid is suited to the description of this model, akin to the use of organoid-on-chip by several contemporaneous models.

2. Line 175 “the mini-bladders exposed to low solute SHU continued to support active proliferation”: The images show that the mini-bladders are not destroyed by the low solute SHU

but they do not show active proliferation (e.g., EdU staining). In fact, if the model accurately mimics the bladder, proliferation should be very low after differentiation and in the absence of any insult.

AU response: We thank the Reviewer for this suggestion. We have performed EdU staining which show the mini-bladders exposed to low solute SHU have a higher proportion of EdU+ proliferating cells than comparable mini-bladders exposed to high solute SHU, as shown in Supplementary Figure 4 D, E in the revised manuscript. For mini-bladders exposed to SHU, the overall proliferation rate is low, and comparable to earlier timepoints of differentiation. We acknowledge that the proliferation potential of the mini-bladder might still be higher than that of the human bladder, as outlined in the sections on Model limitations.

3. Line 244: “which can be explained by improved access to deeper layers of the tissue caused by the loss of tight junctions...”. The authors do not demonstrate whether the loss of tight junctions is directly related. They could either provide evidence (e.g., loosening tight junctions with EDTA in the presence of low or mild solute) or adjust the wording to indicate that it "could" be explained by this mechanism.

AU response: We thank the Reviewer for this constructive feedback which has strengthened this aspect of the manuscript considerably. We have performed the suggested experiment and the data in Supplementary Figure 7E and 2G, H in the revised manuscript. The immunostaining shows lower expression of ZO-1 in mini-bladders exposed to high-solute urine which is corroborated functionally with increased leakage of dextran using the dextran diffusion assay (Figure 2G, H). Further we show that the weakened junctions in mini-bladders pre-treated with EDTA directly correlate with an increased burden of tissue-associated bacteria (Supplementary Figure 7E).

4. In Fig. 4: the authors state that when infecting with high or low solute SHU, 6 hpi is already sufficient to observe differences in tissue-associated bacteria. Do tight junctions disrupt that quickly? Only nuclei counts are shown in Fig. 2 (for SHU) and Fig. S2 (for PHU). Given the importance the authors place on tight junction integrity throughout the manuscript, it would be helpful to include ZO-1 staining over time to better assess these changes.

AU response: We thank the Reviewer for this constructive suggestion, a representative 3D image of the mini-bladder with ZO-1 labelled via immunofluorescence in high and low solute SHU is included in Figure 2G in the revised manuscript.

5. It is unusual that the mini-bladders were grown in different media depending on their appearance. The authors mentioned that in about 10% of the chips, they had to supplement the medium with growth factors (Y-27632, A83-01, CHIR99021, B27, FGF7, 10, and 2). This could have significant effects on the tissue or its response, as for example, Y-27632 and A83-01 are inhibitors of key host pathways. It may also explain why some replicates appear very different from others in some experiments. Defective mini-bladders should be excluded from the analysis.

AU response: We sincerely apologize for the confusion. Model development is a continuous process, and we inadvertently included data from experiments where a different set of protocols were being tested, which could accelerate differentiation. There was no intention to suggest, nor

should it be implied that the protocol used for the vast majority of mini-bladders in the paper generated inconsistent or defective mini-bladders from which data was used in the paper. To clarify this point completely we have replicated and replaced all the data in the original submission that was generated with these mini-bladders. These datapoints were a small subset of datapoints in Supplementary Figure S7B, C, D, Supplementary Figure 9 D, E, Supplementary Figure 4B, C and Figure 3 D, F and G in the revised manuscript. Importantly, the new data agree with the previous versions of these panels in the original submission.

Minor comments:

6. Fig.1. (C,E,G):A close-up in the figures would improve visualization.

AU response: We thank the Reviewer for the constructive feedback. The required closeups are now provided in Supplementary Figure 2 in the revised manuscript.

7. Since the quantification is image-based for tissue-associated bacteria, it would be helpful to include some close-up images. Were these quantified from images similar to those in Fig. 3, 4, S5, and S6?

AU response: We confirm that the quantification is from images similar to Figures 3, 4, Supplementary Fig 6 and Supplementary Fig 7. We show the general workflow for analysis of tissue-associated bacteria in Imaris in Supplementary Figure 10 in the revised manuscript.

8. It would be helpful to see the DRAQ7 staining in the mini-bladders exposed to the PHU (Fig. 2S) to compare it with those in Fig. 2 with SHU.

AU response: We have included representative images for DRAQ7 staining in PHU-exposed mini-bladders in Supplementary Figure 4 B and C in the revised manuscript; the images confirm that outcomes with PHU are along the same lines as SHU.

9. I am sure there is a reason for using SHU or PHU in different experiments, but it is not always clearly stated in the text. Some clarification would help the reader understand.

AU response: We apologize for the confusion. Our experimental design has used PHU to establish the chronic infection model. We then used the consistency and reliability of SHU for further experiments to obtain mechanistic insight and characterisation for different functional readouts. SHU also provides us a means to benchmark the mini-bladder to other in vitro models, as well as provides an experimental approach where the bacteria growth rate is controlled and the same across different conditions. To clearly indicate this to the reader, we have included a sentence in **Line 184-185** of the revised manuscript:

“The two formulations are termed high solute and low solute SHU respectively and are used for experiments in the rest of the manuscript unless otherwise described.”

10. Does the permeability of the model change with exposure to high-solute PHU or SHU?

AU response: We confirm that the exposure to high-solute SHU increased permeability in the dextran diffusion assay. Immunofluorescence imaging confirmed that this was accompanied by a reduction in tight junctions. We have incorporated this data in Figure 2G and H in the revised manuscript.

Figure R1: Permeability measurements via a Dextran diffusion assay.

We observed that high solute PHU also increased permeability significantly relative to low solute PHU (see Figure R1 for Reviewers).

11. It is interesting that while the bacterial doubling time in SHU remains the same between high and low solute conditions, in PHU, there is a significant difference between high and low solute but not as much between mild and high. Could this be due to a lack of nutrients in the low (more diluted) solute PHU?

AU response: We agree with the Reviewer that there is far more heterogeneity of CFT073 growth in PHU as compared to SHU. A complete analysis of the difference between the PHUs is beyond the scope of the manuscript. We are not sure if this would be informative, as they come from different individuals, with different metabolic statuses etc. In fact, it is to overcome these inherent differences in urine composition that we have used SHU as a standard formulation. The SHU formulations are devised to maintain relatively the same nutrient content between high and low solute formulations so that the bacterial doubling time is consistent. This helps in standardising experimental protocols in relation to infection across all the conditions.

12. Representative images corresponding to the measurements in Fig. S6A-B would be good as well.

AU response: We have provided these in Supplementary Figure 8 C in the revised manuscript. Although the bacteria were not imaged in high-resolution at the single-cell level, the significantly larger spherical shapes of the CWD are easily appreciated.

13. Reference 46 is missing.

AU response: We apologise for this omission; the correct reference has now been provided.

Reviewer #2 and 3 (Remarks to the Author)

The paper from Paduthol and colleagues presents an intriguing new human urothelial model, which is used to explore several interesting biological questions including the effect of urine concentration on tissue damage and antibiotic response. It's a tremendous achievement to manage to get (mostly) differentiated, urine-tolerant urothelium, flow and stretch all into one model, which is a great advance from this lab's previous work with Emulate. Overall, the model has potential for discovering some worthwhile new host/pathogen biology beyond what has been explored here, and is impactful for the field. There are, however, some concerns, both major and minor, which we've outlined below in the hopes that these comments might prove useful for improving the manuscript.

AU response: We thank the Reviewers for this very positive evaluation of this work, highlighting the significant improvement of this model over the previous models established in our lab, and suggestions for improvement.

Major concerns

1. The premise of the work is based on literature which the authors use to lay out as a general fact that high-solute urine damages the urothelium ("gradual tissue death", tight junction depletion, apoptosis etc.). I was surprised to see this as I was not familiar with such facts, especially as it seems counterintuitive that any sort of urine would gradually kill the very organ that has been highly evolved and equipped to deal with it. Dehydration would have been a common occurrence in the human race (and all mammals) over millennia, and still is. I looked up each of the papers supporting the idea that urine degrades the bladder (13,14,15,16 and 43). Three of the studies were in rats, one was in rat cell culture, and the last was in toads. Cell cultures cannot tolerate urine, not having the requisite protection, so this may not be the best model system to study cytopathic effects of urine. I'm not familiar with the toad urothelium, but it's well known that rodents have more concentrated urine than humans, and of course they have thinner urothelia, which might make them more delicate. I think it's misleading to lay out these "facts" without acknowledging that they are done in model systems that might not be as relevant to the human bladder – the way it is phrased, the reader may assume these are human studies. I certainly did before checking the references.

AU response: We thank the Reviewers for this feedback and regret that we have generated unnecessary confusion. There was no intention on our part to mislead with regard to references to past literature – most research in infectious diseases is performed in animal models as in place of studies on humans and often cited as mechanistic fact. In the revised manuscript, we have edited **line 45** to start as follows:

Line 45: " In experiments in animal models including rodents, ..."

Similarly, we have edited **Line 389-390** in the Discussion as follows:

Line 389: " ..confirming reports in vivo in animal models "

Our claim is not that high-solute urine gradually kills the bladder in humans. Rather, we believe it has a deleterious effect on bladder resilience, and long-term function particularly in the context of certain stress or injury. These observations are exacerbated in the context and timeline of the mini-bladder model. We fully acknowledge that our model is not a complete recapitulation of the

in vivo bladder, nor does it have a similar longevity, as highlighted in the section on ‘Limitations of this study.’

2. Line 117. “The resultant tissue at day 20 recreates the organization of the human bladder epithelium.” Authors are entirely silent, both here in the Results and later in the Discussion, on the fact that the normal urothelium does not have stretches of undifferentiated, CK20-negative tissue punctuated by blooms of differentiated (possibly hypertrophic?) tissue. The human urothelium should have a uniform layer of intermediate cells 5-6 layers deep. In the example shown in Fig 1, it looks like this ranges from 0 to 5 intermediate cells depending on location. Don’t get me wrong; this model is still an impressive achievement, but its limitations need to be acknowledged up front in the results text and not waved away under the guise of a perfect “recreation”. The fact that a significant subset of the bladder urothelia is undifferentiated also makes us take the high-solute damage with a grain of salt – no pun intended. Maybe these undifferentiated zones are more prone to infection as well.

Given the above, please qualify all sentences such as that in line 151-2: “Together, these results confirm that the mini-bladder microtissue model contains a stratified, differentiated urothelium [with the implication that this is 100% true across the tissue]...” See also Discussion, line 344.

AU response: The Reviewers raise several complex points. First, we agree entirely with the Reviewers that the normal urothelium does not have undifferentiated stretches. We respectfully disagree with the interpretation that this is the case for our mini-bladders. We attempted to show an entire 3D volume for the differentiated mini-bladder in one image- capturing a millimetre or more in X and Y, and nearly 300 μm in Z in one image, or a volume of $3 \times 10^8 \mu\text{m}^3$. This necessitated contrast adjustments that make it seem like areas with relatively lower intensity of CK20 staining look like there have no CK20 staining. In Supplementary Figure 2A in the revised manuscript, we have changed the intensity scaling to show that even in these areas, there is staining for CK20, and in a similar manner, we also show that there are no large “gaps” or “holes” in the CK13 staining. To further convince the reviewers we have provided magnified images of sections of the urothelium with the same resolution and spatial size as the current state-of-art shown with Transwells in Supplementary Figures 2C-H in the revised manuscript. In this case, the epithelium appears locally flat, and the stratification is clearly visible. Thus, we find no evidence for the conclusion that a “significant subset of the bladder urothelial is undifferentiated,” or the subsequent extrapolations of this claim by the Reviewers. Indeed, all the images in subsequent figures clearly show bacteria throughout the bladder, which would be unlikely if infection was concentrated at the “undifferentiated zones” as claimed by the Reviewers.

We acknowledge that the epithelial surface is more undulating than similar models based on a flat architecture grown on rigid plastic substrates e.g., Transwells. We believe this is likely more reflective of the real 3-D mechanoactive organ ¹. Evidence for the uniform nature of the stratification in Transwell models itself is also only provided in the literature with comparatively high-resolution images. In the mini-bladder model there are regions with a higher number of intermediate cell layers and those with a thinner urothelium, a difference which can be easily obscured if one showed higher resolution images. Overall, the limitations and avenues for further development have been outlined in the paragraph on limitations of the study.

3. Line 162. Experiments are said to be done here with mini-bladders that had “differentiated uniformly”: this is interesting, as it confirms that perhaps they often don’t, as demonstrated in Figure 1 – but we should be shown an image or two of fully/uniformly differentiated and stratified bladders in Figure 1 as well, just so we can see for ourselves. By extension, are we to assume that all the other experiments were done with the incompletely developed mini-bladders shown in Figure 1?

AU response: We regret a loosely worded sentence but also the connotations derived from them that have been extrapolated to the rest of the manuscript. What we meant to report was that the devices were differentiated under exactly the same conditions starting with the same batch of initial cells, to minimize batch-to-batch variability. The sentence has now been edited for clarity in this regard. No data in the original submission or the revised submission was obtained from experiments performed with incompletely developed mini-bladders.

4. Fig 1K. Knowing whether fusiform vesicles are present in the model, meaning that the recycling that has been shown to allow bacterial entry during the stretch/relax cycle (e.g. Truschel et al.) might take place, is crucial to understand how physiological your dynamic bladder model is. We as readers cannot be satisfied with Figure 1K without seeing any images of these vesicles. There is nothing evident in Figure 1J – not surprising at that resolution. Representative images of the correct resolution that were used to quantify these vesicles **absolutely must be shown** - in Supplementary is fine. At such resolution, it would also be possible to see hinges/asymmetric unit plaques by EM and this would also be very useful to confirm physiological relevance – as you are showing evidence of damage from high-solute urine, one could argue that this isn’t surprising if the model lacks AUM plaques. The authors should also confirm the presence of the uroplakin by IFA – it is a straightforward thing to check as there are a number of great antibodies. As the authors will know, the uroplakin complex will not form properly unless all components are present and correct – only UP3 RNA was tested. The question of whether intact uroplakin complexes are elaborated on the surface and make AUM is arguably more important for infection studies and differentiation status than is CK20. If you tried to see UP3 or other UPs by IFA and the model does not express them on the surface, this needs to be admitted. We aren’t shown any higher resolution top-down confocals with actin either – do you see the expected enlarged, honeycomb-like pattern? Bottom line, are these apical cells truly terminally differentiated umbrella cells or not?

AU response: The Reviewers have raised several interesting points; we thank them for this constructive feedback.

Fusiform vesicles: We now provide higher resolution images of the unstretched and stretched tissues in Supplementary Figure 3 in the revised manuscript and representative zooms of the vesicles in each case are also shown. Information on how the number of vesicles were counted is provided in the section titled ‘Analysis of Vesicles’ in the Materials and Methods.

Hinges/plaques: These phenomena are challenging to see. In the revised manuscript (Supplementary Figure 3 C, D), we highlight areas of intense contrast in SEM images of the tissue that is localised to the very apical edge of the junction. This extends for a short distance along the surface of the plasma membrane, providing the characteristic curve on the membrane expected when viewing the hinges/asymmetric unit plaques side-on. However, we recognize that the mini-

bladder does not recapitulate this feature to the same level and extent as in vivo, and this is an area for further development. This is outlined in the paragraph on limitations of this study in the Discussion.

Uroplakin immunostaining: We provide evidence of strong expression of UP3A and UP1A on the apical surface of the umbrella cells in the mini-bladder model in Supplementary Figure 2D, E in the revised manuscript.

Umbrella cell morphology – there are several panels that reflect the umbrella cell morphology. This is evident in the immunostaining in Figure 1D, which shows binucleate larger cells facing the lumen of the mini-bladder. Additional examples of binucleated cells are shown in Supplementary Figure 2B where the honeycomb-like Actin pattern is also evident. Lastly, the binucleated cell in Figure 5J is also larger than the cells below in in the same Z-stack.

5. Regarding the solute status in human and synthetic urine – matching osmolarity seemed to be your only focus. But as I understand it, the kidneys concentrate different solutes differently; for example, urea is subject to disproportionate concentration compared with other chemicals. Just diluting globally to the right osmolarity implies that it is only osmolarity per se that governs any downstream effects, when actually it might be the concentration of one or a few key chemicals, which again are not proportionally concentrated by dehydration. I think getting the proportions right is crucial – which could be found in the literature, or by analyzing the components of the human urine you are using, and making informed choices how to make up the synthetic high and low urines. If my statements are incorrect, perhaps insert a line in the paper about how your choices were made.

AU response: We thank the Reviewers for this comment. Our goal in this study was not to invent a better formulation of synthetic urine; instead, we relied on the published literature that has established a formulation that has been used by several labs in a clinical setting and specifically for urinary tract infections (e.g., Abbot et al ²). The media has shown to support the growth of both gram-positives and gram-negatives; thus, it was important to demonstrate that our model is compatible with this media when administered in the formulation reported by Ipe et al. 2016 ³. We have edited the sentence **at line 173** to further clarify our intent.

Line 173: Ipe et al developed a protocol for synthetic human urine (SHU) that closely mimics human urine in nutritional content but with well-defined ingredients. This formulation has been shown to support the growth of several uropathogens and has been used as a surrogate to pooled human urine to model antibiotic responses in urinary tract infections.

In response to query 8 from Reviewer 4, we also provide evidence of the formation of filaments in SHU in the conditions used for the experiments. This is a further validation of this formulation. We agree that the exact solute compositions that might contribute to osmolarity driven changes to a greater or lesser degree is a matter for further investigation as is the development of the next-generation of synthetic urines.

6. In a related vein, pH is known to affect bacterial behavior as well as drug action. Why have

you made your synthetic urines so acidic compared with the urine you've collected from volunteers?

AU response: We thank the Reviewers for raising this important point. We regret that in the first submission, a crucial methodological detail was inadvertently omitted from the description in the Materials and Methods. While we do generate SHU formulations at pH = 5.6, this is to follow the established protocol. The acidic pH ensures long-term stability as the formulation tends to generate precipitates over time. However, when introduced into the mini-bladder, the pH of the SHU was adjusted by addition of 20 mM of HEPES buffer. This effectively changes the pH of the SHU to above 6, which is closer to that of the PHUs collected (updated in Supplementary Table 1). Thus, we do not think the pH mismatch is as severe as it might appear. The Materials and Methods and Supplementary Table 2 (SHU formulation) have been updated to reflect this.

7. Limitations. In the discussion, it should be acknowledged that, from what is shown, the mini-bladder might not be uniformly stratified and differentiated in some/many experiments (depending on how often they look like Figure 1), and this state could absolutely affect how easily the tissue could be damaged by high solutes, as well as susceptibility to infection. It should be clear to the reader that you understand the limits of the model, and where there is room for future improvement.

AU response: We disagree with the assessment that the images in Figure 1 represent mini-bladders without a very high degree of differentiation, and stratification (albeit variable). Nevertheless, we fully agree that our model still has limitations, and room for future improvement, including related to stratification and differentiation. We have elaborated on both points in the section on “Limitations of this Study” in the Discussion, and we hope that the Reviewers will find this satisfactory.

Minor points

1. What sex are H-6215 primary human bladder epithelial cells? This should be specified in the Methods.

AU response: The necessary details have been added to the revised manuscript.

2. Line 687. “In case of any tissue defects or failure of stratification, post day 10 the basal media was supplemented with propagation factors...” This is a massive protocol difference between cultures with and without this propagation fudge factor. Experiments with one or other should be noted in the figures as the two types of tissue may not be comparable in all aspects.

AU response: We are grateful to the Reviewers for pointing out this difference and apologise for the confusion. We draw the attention of the Reviewers to the Response to point 5 by Reviewer 1 who raises the same issue. In the revised manuscript, the sentences have been removed as all mini-bladders were differentiated with the same protocol. Experiments have been repeated where needed to provide the necessary datapoints. In no case have any of the conclusions changed.

3. What is the rationale for using high solute urine for differentiation? Sorry if I missed it.

AU response: We chose to use high solute SHU as it is the published formulation for consistency of protocol and to generate a formulation that could be reproducible across labs. The inability to use undiluted urine was also a clear limiting factor in our earlier bladder-chip model (Sharma et al. ⁴) and so this motivated us to demonstrate the ability to generate a fully stratified tissue in undiluted urine as we do in this study.

4. MOI describes the probability of initial collision with surface host cells (Poisson distribution). Better to calculate MOI based on the estimated superficial cell number across the accessible surface area. Or at least give both figures so the reader can compare it to conditions in their own infection models.

AU response: The MOI is now mentioned at **Line 222** in the Main Text of the revised manuscript. One caveat we would like to highlight is that it is more difficult to accurately determine the number of host cells available for attachment which would be the superficial layer and whether all parts of the 3D tissue would be equally exposed.

5. Fig 1A. Add some direction of flow arrows into schematic just to make schematic clearer.

AU response: We thank the Reviewers for this suggestion; this has been implemented in Figure 1A in the revised manuscript.

6. Fig 1G. ZO-1 is located at cell junctions. Hard to know if ZO-1 is expressed in between cells at this resolution. Could the authors use a higher objective for greater cell resolution and if it is logistically possible, image from top-down?

AU response: High resolution imaging is more challenging in this model, and if the Reviewers mean from top-down that we should use an upright microscope, then this is not experimentally feasible. Nevertheless, we have generated higher-resolution images of sections of mini-bladder that were accessible via the inverted microscopy, and this is shown in Supplementary Figure 2 in the revised manuscript. The presence of ZO-1 in the junctions is visible in Supplementary Figure 2H.

7. Fig. 1H & I. Is this from stretched or unstretched samples? And would this change with the physiological difference?

AU response: The data in these figures is from the unstretched conditions. Carattino et al 2013 ⁴ showed that the stretching of ex-vivo rabbit urothelium mounted in Ussing chambers reduced TEER but not the permeability of fluorescein, concluding that stretching increased ion conduction without creating a leaky tissue. We therefore would expect no change in our model with the dextran assay due to its functional similarity to the fluorescein assay. However, it was not possible to perform these measurements at the frequency and consistency needed for incorporation into a regular protocol.

8. Fig 1F. The assay used to quantify “expression” is not mentioned in results text or figure legends. Readers should not have to trawl through the methods to find out if this is protein level (image quantification) or RNA. I couldn’t find it in the methods either. As there is some RNA

work in the paper, and there are no IF images of the parental cell line, perhaps it's RNA expression? Entirely unclear. I may have just missed it, but figure legends should tell you what sort of assay it is at the bare minimum. "RE units" on the Y axis could be any sort of arbitrary unit including fluorescence intensity so they don't shed light.

AU response: The Figure Legend has been updated to state that RE refers to relative expression and that the assay used is qRT-PCR.

9. 131. Nit-picky point perhaps, but strong ZO-1 staining does not in fact confirm that the tissue retains a 'robust barrier'. Only the subsequent functional assay can.

AU response: We agree with the Reviewers on this point and therefore have provided functional readout on the dextran diffusion assay wherever relevant including in Figure 1H, I, Figure 3F, G and now additionally in Figure 2H in the revised manuscript.

10. Why didn't the authors do TEER? If the model is accessible to the probe, that would have been interesting, to compare with data from other models.

AU response: TEER is easier to implement in more accessible geometries such as the Transwell but more difficult to implement on a consistent basis in the mini-bladder. We feel the dextran diffusion assay is a highly relevant alternative assay used by several publications.

11. Fig 1J. The unstretched bladder EM looks very reminiscent of bladder biopsies. But in the stretched version, I'm not really seeing the flattened umbrella cells I'd expect. The image is very limited (2-3 superficial cells). Could further examples be presented (even if in Supplementary) so we can view the 'average' superficial cell morphology? This goes back to major point 4, above.

AU response: We thank the Reviewers for the positive evaluation of the EM data from the unstretched bladder, particularly in that they are reminiscent of bladder biopsies. Additional EM images of the mini-bladder in unstretched and stretched conditions are included in Supplementary Figure 3A,B in the revised manuscript and show that the umbrella layer has a flattened morphology.

12. All your tables are supplementary but in the text they are referred to as Table 1, Table 2 etc., not S1, S2, which sent me on a fruitless hunt.

AU response: This has been corrected in the revised manuscript.

13. Figure 2. Given that an infected urothelium will shed cells, can your bulk transcriptomics distinguish between "down-regulation" of proteins vs loss through absence of cells? All your effects are described as downregulation. Line 181-2 – 'Long-term exposure to high solute SHU led to lower expression of cell-cell adhesion and proliferation gene sets (Fig 2E)'. Sorry if I'm missing something, but if high solute PHU is important for proliferation of immature bladders then how can SHU high /low solute then lead to lower/higher expression cell adhesion and proliferation, respectively?

AU response: We thank the Reviewers for raising several interesting points. The RNAseq was done in the absence of any infection, therefore we do not expect a large incidence of shedding. Nevertheless, we cannot rule out that the effect we see at the transcriptional level is indicative of the loss of certain cell types. This is likely to be particularly the case for cell-type specific markers mentioned at **Line 197**. In contrast, changes in more general markers of tight junctions

are likely reflective of overall changes in this tissue. We have therefore nuanced the sentence on cell-type specific markers in the revised manuscript as follows:

Line 197: In particular, genes related to urothelial differentiation such as *KRT8*, *KRT5*, *KRT20*, *KRT13* and *UPK2* had lower expression in conditions with long-term exposure to high solute SHU (Fig 2F)...

Our protocol relies on urine for differentiation, and to maintain a consistent protocol we used the SHU formulation developed by Ipe et al., which has a high osmolality. We do not claim that high solute PHU is a necessity for bladder differentiation. Further studies will evaluate the optimal osmolality conditions for bladder differentiation.

14. Line 201-202 – Is there difference to CFT073 if pre-exposed to low and high solute urine?

AU response: We thank the Reviewers for raising an interesting point. As we outline in Supplementary Table 1, growth in urine of different compositions is varied and so therefore for ease of experimentation and for compatibility with previous protocols we did not pre-expose the bacteria to urine. Additionally, growth in static LB to stationary phase has been shown to enhance type I pili production, and was used in the bladder organoid and bladder-chip models developed in the lab, thus we retained this protocol to be able to compare the performance of this model with those. Nevertheless, while this may change the kinetics of the very early stages of infection, it is unlikely to impact the overall time course or outcomes of the experiment. This may be observed in the chronic micturition experiments where the bacteria that reseed the infection in the bladder after micturition are those that have been exposed to urine in the mini-bladder for several hours. Overall, this is a useful methodological innovation that will be examined in future studies.

15. Line 208 – IBC formation. Could Figure S4E show an orthogonal view to show depth of IBCs?

AU response: We have included an orthogonal view for this panel; it is now in Supplementary Figure 6E in the revised manuscript.

16. Figure 3B – Please confirm laser power for confocal microscopy is the same for all images?

AU response: We confirm that the laser power used was the same for all images in this panel. Further details on image acquisition and analysis workflow are incorporated in the Methods section and the new Supplementary Figure 10 in the revised manuscript.

17. Line 215. “but washing did not completely clear the lumen. This is not unexpected as bacteria near the epithelial layer boundary experience lower urine flow rates...” The context around this point could be more nuanced given that flow has actually been shown to increase UPEC adhesion, which isn’t mentioned here.

AU response: We thank the Reviewers for this comment and agree that there is the possibility of adhesion from type I pili activity that was not mentioned. We have rephrased the sentence as follows in the Revised Manuscript:

Line 236 - This is not unexpected as bacteria near the epithelial layer boundary experience lower urine flow rates, and may also adhere better to the umbrella cells because of the catch-bond mechanism of type I pili adhesion.

18. The word ‘axenic’ is not perhaps the best choice, as it is usually used to reference single-species growth in comparison to polymicrobial growth with other bacteria or parasites, which is not studied in this paper. I took a survey on my floor and only about 2 people out of 20 had even ever seen this word before. Most people in the field would say broth culture?

AU response: We thank the Reviewers for this feedback. In this manuscript we culture UPEC in LB broth, in synthetic urine in a flask, and in urine in the mini-bladders. The use of this term allows us to refer to conditions of culture of microbe only, separate from the co-culture with mammalian cells. This fits with the definition of the term and we therefore will retain it.

19. Line 289. Technically speaking, a still image cannot distinguish whether a bacterium is going from coccus to bacillus or vice versa.

AU response: We agree with the Reviewers, but in this case we do know that the bacteria were rod-shaped before the Fosfomycin treatment and so we can infer the direction of change. We further wish to clarify in case there is a doubt that in these experiments, the rod-shaped bacteria are transitioning to a cell-wall-deficient form (the delaminating cell wall is clearly visible) and not coccoid shapes.

20. Line 292. Here or in the discussion, you should explore the relationship between the CWD spherical E. coli you saw, versus the coccus-shaped E. coli found in mouse IBCs. Do you think these are totally different beasts?

AU response: We thank the Reviewers for raising this point. The CWD bacteria are very different from the coccoid shaped bacteria in IBCs, which clearly retain their cell wall. Notably, the CWD bacteria are swollen and larger than rod-shaped bacteria, whereas the coccoid shaped bacteria in IBCs are usually smaller in size. To clarify this, we have edited the sentence accordingly in the revised manuscript:

Line 329: “The latter population had cell sizes consistent with the morphology of CWD UPEC in the urine, and not the densely-packed smaller coccoid bacteria observed in IBCs.”

21. Line 353. Seems to imply that hydrated individuals would not get UTIs. That is likely not true: rephrase. Also begs the question why rodents, with their more concentrated urines, are difficult to give a UTI. Perhaps this should be discussed.

AU response: We disagree with this interpretation. Rather, we think that prolonged exposure to concentrated urine or dehydrated conditions weakens bladder resilience, which could contribute to susceptibility to chronic infection. Rodent urine differs significantly from that of humans and so we do not think these results can be extrapolated to rodents. Interspecies

differences such as these are well-documented but difficult to disentangle given that rodents are often used as models for humans. Further work with models such as the mini-bladder can disentangle these.

22. Is there a difference between stretched and unstretched bladder with regard to infection? Maybe more clarity needed in text whether differences are seen with this phenotype.

AU response: The authors raise an interesting question that we are pursuing in follow-up studies. In this manuscript, the stretching experiments were performed to showcase the capabilities of this model, but the long-term impact of stretch in the context of infection was not the focus of this work. However, we do point to earlier work from our lab (Sharma et al., eLife 2021⁵) which showed that urothelial bacterial burden was higher in bladder-chips undergoing periodic mechanical stretch which indicates that mechanical forces are contributing factors to infection.

23. Figure 4 - Is 2.5 hours enough to show regrowth with ciprofloxacin? For Fosfomycin, in low solute urine would regrowth rebound to higher CFU levels if more time for regrowth were allowed?

AU response: We are somewhat puzzled by the comment regarding Ciprofloxacin, as in Figure 4 we measure regrowth for 9 hours after withdrawal of antibiotic. Regarding Fosfomycin, if maintained long enough growth in low solute urine would be expected to rebound as rod shaped bacteria proliferate to the same extent in the mini-bladder in both urine formulations.

24. Line 281 – Will D-sucrose or BHI affect growth by introduction of new carbon sources, which may explain higher urinal CFUs in figure S6C & D?

AU response: We do not observe an appreciable difference in CFU for exponentially growing UPEC in axenic culture plated on LM or LB, indicating that the different nutritive content between the two formulations is not the reason for higher recoverable. This data is shown in Supplementary Fig 8D in the revised manuscript.

25. Would you find higher CFU counts 29 hpi in tissue associated CFUs as CWD may have invaded into bladder? Only urinal UPEC tested (Figure S6 E & F).

AU response: We do not measure tissue-associated CFUs in this manuscript as this is an endpoint measurement and is not suited to capturing the kinetics of growth and invasion simultaneously. However, the data in Figure 4G shows clearly an increased in the tissue associated bacteria by 29 hpi when quantified by bacterial fluorescence.

26. Under what ethical framework were patients consented for urine? Please give IRB/ethics committee and license number in the Methods as is standard, or insert a statement why this is not necessary (perhaps they do things differently in Switzerland than in the rest of the world?)

AU response: We apologise for the omission of this information in the original submission; this is now included in appropriate section in the Methods in the Revised manuscript.

Reviewer #4 (Remarks to the Author)

In this study Paduthol et al. have developed a urothelial organoid model to study tissue resilience and bacterial recurrence during model UTI. Building on previous work from this group, and others, in this study the authors have combined various important aspects of the infection cycle from a host perspective – e.g. stratification and dynamics of micturition – in a single system. Overall, I find this work to be of technically high standards with complementary approaches used, the biological leaps are perhaps a bit less advanced but would likely be interesting for an audience relevant to Nature Communications. Apart from a few general issues (outlined below), I think this study is a significant step forward in the infection-on-a-chip model development and could be accepted, pending satisfactory answers to the comments.

AU Response: We thank the Reviewer for the very positive assessment and recommendation for acceptance and constructive feedback that has improved the manuscript and increased its impact.

General/model comments:

1. The figures are quite busy, and figure legends not always clear to follow. Could this be looked over throughout.

AU response: We thank the Reviewer for this constructive feedback. We have edited the figure legends throughout to make them easier to follow and to add more information where needed.

2. The superficial layer(s?) of cells seem quite uneven (Figure 1C,E,G), how did that correlate with bacterial invasion? Did the authors notice if the green blobs (multiple IBCs? Figure 3B) colocalised with these protrusions of bladder cells?

AU response: The Reviewer raises an interesting point. It is true that there are sections of the mini-bladder that have greater stratification than others, however as discussed in our response to query 2 from Reviewers 2 and 3, the epithelium is still locally flat. We did not observe a correlation between the protrusions and increased infection.

3. One possibly important issue is that the UPEC strain CFT073 is used, this strain could be viewed as substandard when it comes to study cystitis (common strain would be UTI89), as CFT073 was isolated from a pyelonephritis patient. I believe this study would be significantly stronger if data from an UTI89 strain was to be included. The experiments should at the very least be repeated with a $\Delta fimH$ (main adhesion protein in bladder infection) or $\Delta papG$ (main adhesion protein in kidney infection) strain, i.e. CFT073 $\Delta papG$, to show that they system behaves like previously shown *in-vivo* systems.

AU response: We thank the Reviewer for this constructive feedback. We have performed key experiments with the UTI89 strain with PHU in the mini-bladder. Our results in Supplementary Figure 6 B, C, D and F in the revised manuscript show that this strain also proliferates in the mini-bladder similar to CFT 073, albeit slower. This is consistent with the greater urothelial damage caused by CFT073 reported in *in vitro* models⁶. A more detailed characterization of the differences between species is beyond the scope of this paper. Data for the $\Delta fimH$ strain is

included in Figure 3 E, G in the revised manuscript which shows that it phenocopies the exogenous addition of D-mannose with reduced tissue-associated bacteria, urinary bacterial burden, and tissue permeability.

4. Line 178: RE: transcriptional profiling of mini-bladder tissue exposed to high- and low-solute SHU. A comparison should be made to the qRT-PCR expression profiles from Fig. 1F. Were relative gene expression comparable for specific targets (e.g., P63 and others)? How would expression profiles differ (especially those related to chemokines and TLRs) between high-solute PHU when compared to high-solute SHU?

AU response: We have included a comparison for qRT-PCR expression profiles in the mini-bladder at D=0 with expression in mini-bladders exposed to low and high solute urine for 6 days Supplementary Figure 5C. The expressions are comparable or in line with the trends discussed. The goal of our study was not to characterise different urine formulations and so the immunological differences between PHU and SHU have not been explored, however in **Figure R2** below expression profiles for TLRs expressed in the human bladder⁷ (*TLR2*, *TLR4*, *TLR5*, *TLR9*) and chemokines like *CXCL1*, *CXCL2* and *CCL2* that drive monocyte and neutrophil recruitment are shown. These show no statistically significant difference between high solute PHU and high solute SHU exposure.

Figure R2: qRT-PCR data for selected genes from mini-bladders under specified experimental conditions (Statistical significance calculated using unpaired t-test)

5. Line 211: Can the authors provide more explanation into the micturition cycling events of the mini-bladder model? Previous models have utilised a urine-flow system which likely also mimics the micturition cycles within UTI, albeit less natural due to the constant flow.

AU response: We apologise that this important feature of the model was not clearly described. We have included additional details in the Materials and Methods of the revised manuscript. Essentially, the lumen of the mini-bladder is accessible via inlet and outlet channels to large reservoirs which can be accessed by a laboratory micropipette. The withdrawal of urine from one reservoir and the addition of urine to the other causes a flow of urine through the mini-bladder until the volumes equalise. These “washes” mimic the voiding and refilling of the bladder with fresh urine.

6. The amount of bacteria seems high (5×10^8 cell/mL, times 3 perfusions), can the authors comment on why they used this high numbers. To me this do not seem like a physiological bacterial load.

AU response: We apologise for the confusion on this point. In our infection protocol for CFT073, ca. 100 μ L of a 5×10^8 cells/mL is added to one reservoir, flowed through the channel, collected from the reservoir, and re-introduced into the original reservoir. After two such rounds, the mini-bladder is considered infected and the infection monitored thereafter. Thus, the mini-bladder is exposed to a total of 5×10^7 bacteria.

7. There is a clear lack of immune cells in the current model. Addition of immune cells to this model would significantly strengthen the study. But I understand that doing so may require a fair amount of extra experiments. Maybe a consideration for the future, but the very least this should be expanded on in the discussion.

AU response: We agree with the Reviewer that the lack of immune cells is a limitation, but also that this is a goal for future studies. We have incorporated this in the section on Limitations of this Study in the Discussion in the revised manuscript.

PHU vs SHU:

8. For flushed bacteria, did the authors find filaments in the SHU experiments? And did they see differences between PHU and SHU (frequency, length, amount ect?), and between high and low solutes? If they saw no filamentation in the SHU, it would be a red flag of the chemical composition as it has been shown in various models (both in-vivo, in-vitro and human) that UPEC undergo morphology changes into a filamentous form during UTI when exposed to real human urine. I'd say it is essential to show that morphology change also in SHU.

AU response: We thank the Reviewer for this important suggestion. In the revised manuscript, we show evidence of filamented urinal UPEC amongst the flushed bacteria in the high and low solute SHU conditions in Supplementary Figure 6G in the revised manuscript. We observed no significant differences between high and low solute SHU conditions. Further characterisation of differences in filamentation between SHU and PHU is beyond the scope of this manuscript.

9. Another critical aspect of this is the growth of UPEC in PHU vs SHU. Others have in the past tried to formulate a SHU that is similar to that of a PHU, with varying success. Notably, if I recall correctly, I don't believe that Ipe et al, did that comparison in their paper (cited by the authors as source for

their SHU). It needs to be shown that bacteria grow at a similar rate in the two, otherwise the comparison is not particularly informative.

In addition, the pHs are different between the PHU and SHU (table S1), this too has been shown to affect UPEC growth and morphology in the past, but is not commented on (only how varying concentrations affect the bladder cells themselves, Lines 167-176).

AU response: We thank the Reviewer for raising important points. In the revised manuscript, we have included data on doubling time for UPEC in Supplementary Table 1 for the different urine formulations used in this study. The doubling time in high solute SHU is consistent with high solute PHU. There are differences between the doubling times in the lower solute conditions. This is partly by design, as the SHU formulations retain the same nutritive content and remove bacterial growth rates as a variable between the conditions. If anything, we expect the lower growth rates in low solute PHU to further accentuate the differences seen between the conditions.

With regards to pH, we apologise for an omission on our part. The SHU formulations are generated as described and stored at pH=5.6 for long-term stability. However, in the experimental setup, the pH was increased to above 6 by the addition of HEPES buffer. This reduces the gap in pH between PHU and SHU formulations quite significantly. The Materials and Methods and Supplementary Table 2 (SHU formulation) have been updated to reflect this.

10. The initial bacterial burden is higher for high solute SHU, so this is perhaps not very surprising that the final bacterial burden is also higher, by the authors interpreted as lower antibiotic clearance (Figure 4D). Would the conclusions change if you for example normalised these data in order to compare?

AU response: The Reviewer has raised a very valid point. We have reanalysed the data as suggested, and the conclusions are unchanged. The plot with normalization is included in Supplementary Figure 7F in the revised manuscript.

11. Line 202: Can the overall density (ρ) of the urine be determined and reported in Table 1? Previous studies have shown that the specific ρ of PHU impacts bacterial cellular fitness and survival within the intracellular bladder environment and may also impact the bladder ecosystem.

AU response: We have incorporated this data in Supplementary Table 1 in the revised manuscript. We see very similar density between all conditions of SHU and PHU.

12. Line 240: Can the authors give reasoning as to why they have used SHU instead of PHU for the antibiotic mini-bladder model assays? I understand that SHU is more robustly made and consistently produced for both high- and low-solute concentrations, but are additional crosstalks between cipro/fosfo treatment and the PHU that may have been missed?

AU response: The Reviewer raises an important point. As noted in Supplementary Table 1, the growth rate of UPEC is variable between the different PHUs, reflective of the donor-to-donor variability. This would make it challenging to perform antibiotic time-kill assays across different

urine formulations. The SHU formulations allow for this. While the kinetics of bacterial clearance may well differ in PHU infections, we do not think there is additional crosstalk that might have been missed. Of note, we do see L-forms generated in UPEC treated with Fosfomycin in axenic culture in high solute PHU. This is shown in Supplementary Figure 8L in the revised manuscript.

CWD:

13. Line 290: Given the fact that CWD UPEC can revert back to normal rod-shaped after antibiotic pressure is removed may suggest changes in gene regulation drive the CWD phenotype in the presence of fosfo. Can the authors consult the literature for transcriptomics studies with acute fosfo treatment in E. coli to better understand this mechanism or alternatively perform mechanistic analyses in their CFT073 model +/- fosfo treatment or in axenic high-solute culture. This will also give more functional insights into the persistence of CWD UPEC within the bladder tissue.

AU response: We thank the Reviewer for raising this important point. The CWD phenotype is a specific response to the mechanism of action of Fosfomycin, which is known to target the very early stages of peptidoglycan synthesis. CWD bacteria, by virtue of the loss of cell wall are no longer targeted by the antibiotic. Work in our lab (PhD thesis of Dr Frédéric Normandeau ⁸) has analysed this response and shown a role for colonic acid biosynthesis pathway (see **Figure R3** below). Further work in this regard is ongoing to understand the link between this pathway and osmolarity driven changes, but this comprehensive study will be the subject of a future report.

[Figure Redacted]

14. Line 318: What is the proportion of CWD and non-CWD UPEC in high-solute SHU given the heterogeneous nature of CWD bacteria?

AU response: We have quantified this through the images shown in Supplementary Figure 9B in the revised manuscript, and this suggests a proportion of CWD: non-CWD of 9:1. We have also quantified the differential colony counts on LB and LM in Supplementary Figure 9C in the revised manuscript that shows the predominance of CWD UPEC in high solute SHU.

Methods:

15. Overall the model and biological methods sections are mostly well explained. There are issues with the imaging sections: they should be more detailed.

In general: not adequate description of fluorescence imaging, for example:

“The sample was imaged using Leica SP8 confocal microscope with a white light laser and 25x water immersion objective (NA = 0.95, Leica).” (Lines 803-804). “Fixed samples (e.g., Fig 1D, 5E, F, S4E) were imaged using Leica SP8 confocal microscope with a 63x water immersion objective (NA =1.20, Leica)” (Lines 828-830). What wavelengths were used, at what laser powers, pixel size, acquisition times, etc? It would be impossible to repeat these experiment based on the information given currently.

AU response: We apologize for the lack of information; this has been duly corrected in the revised manuscript. The relevant details are now included in the Materials and Methods section of the revised manuscript.

16. The images in Figures 3B, 4B and 4E are of varying quality. Particularly the fluorescence channel, it is hard to truly assess the data based on these images, was the fluorescence in the full volume of the images superimposed? I’m not questioning that the signal increased over time, looking at the images it does look like that qualitatively. But it is not immediately clear how quantification was done when comparing end point fluorescence. There is no description of how they were acquired nor analyzed (or I couldn’t find it at least. Lines 823-824 is not a well enough description of the analysis). Images like Figure 1D, including bacteria, clearly showing that they are intracellular show be included. Otherwise, how can the authors claim that the schematics only based on this model? This is also an excellent opportunity to show “how deep” the UPEC travels in the stratified layers.

AU response: We regret that the original submission did not provide sufficient methodological details for the image analysis. This has been corrected in the revised manuscript. The revised manuscript also includes a Supplementary Figure 10 outlining the image analysis pipeline followed in Imaris. In each case, we analysed a central volume of 650 μm x 650 μm x 100 μm cuboid in the mini-bladder, and quantified parameters based on 3D analysis. While we appreciate that this is not easy to grasp in the images in the Main Figures due to the lack of space,

we hope that the higher resolution images of intracellular bacteria in Supplementary Figure 6E, F, the new images of IBCs in Supplementary Figure 6E, the new data on the $\Delta fimH$ mutants and the D-mannose experiments in Figure 3, both of which show clearly visible reductions in bacteria and the volumetric EM in Figure 5 are convincing as to the intracellular nature of the bacteria.

References:

1. Zhou, H., Guo, C. C. & Ro, J. Y. *Urinary Bladder Pathology*.
2. Abbott, I. J. *et al.* Evaluation of pooled human urine and synthetic alternatives in a dynamic bladder infection in vitro model simulating oral fosfomycin therapy. *J Microbiol Methods* **171**, (2020).
3. Ipe, D. S. & Ulett, G. C. Evaluation of the in vitro growth of urinary tract infection-causing gram-negative and gram-positive bacteria in a proposed synthetic human urine (SHU) medium. *J Microbiol Methods* **127**, 164–171 (2016).
4. Carattino, M. D. *et al.* Bladder filling and voiding affect umbrella cell tight junction organization and function. *Am J Physiol Renal Physiol* **305**, 1158–1168 (2013).
5. Sharma, K. *et al.* Dynamic persistence of intracellular bacterial communities of uropathogenic escherichia coli in a human bladder-chip model of urinary tract infections. *Elife* **10**, (2021).
6. Flores, C. *et al.* *A Human Urothelial Microtissue Model Reveals Shared Colonization and Survival Strategies between Uropathogens and Commensals*. <https://www.science.org> (2023).
7. Jafari, N. V. & Rohn, J. L. The urothelium: a multi-faceted barrier against a harsh environment. *Mucosal Immunol* <https://doi.org/10.1038/s41385-022-00565-0> (2022) doi:10.1038/s41385-022-00565-0.
8. Normandeau Frédéric. Exploring the Role of Unstable L-Forms in the Survival of Uropathogenic Escherichia coli Under Fosfomycin Exposure Frédéric NORMANDEAU. (EPFL, Lausanne, 2025).

Point-by-point reply

We thank the reviewers for their positive comments on and enthusiasm for the revised manuscript and are pleased that they all recommend publication. Below we address the remaining outstanding points raised by Reviewer 1.

Reviewer #1 (Remarks to the Author):

The authors have made a significant effort to address the comments, and the manuscript has improved substantially. Most of my comments have been addressed appropriately, although I still have concerns with the following points:

- 1. Related to Major Comment 1: I continue to disagree with the authors. The fact that other studies have used the term organoid incorrectly does not justify its use here. While organoids can indeed be derived from primary cells, these organoids are grown in extracellular matrix in a three-dimensional context, which was not done here. Instead, the authors describe culture conditions that rely on standard cell culture medium and passaging with TrypLE Express. This is consistent with conventional primary cell culture rather than organoid culture. Therefore, the terminology should be corrected, and the term organoid should be removed from the title.**

AU response: We politely disagree with this restricted interpretation of the term organoid and characterization of the model as an extension of standard cell culture. In our model, cells are also grown in a three-dimensional extracellular matrix architecture. Nevertheless, we have edited the title for brevity and removed the term organoid.

- 2. Related to Major Comment 2: Supplementary Figure 4D does not show a representative image of the control (D0), which is the condition shown in the corresponding graph (nice data btw).**

AU response: Thank you for the positive feedback, the necessary representative image has been added to Supplementary Fig. 4D in the revised manuscript.

- 3. Related to Major Comments 3 and 4: ZO1 staining is expected to localize at cell-to-cell junctions, not across the entire apical surface. In the new figure panels (Figure 2G and Supplementary Figure 2H), the ZO1 signal appears to extend across the whole apical membrane. Image resolution does not explain this observation, since the resolution should be comparable to that of the phalloidin images (Supplementary Figure 2B). The authors should clarify how they interpret this staining pattern.**

AU response: The image shown for ZO-1 is a maximum intensity projection of the entire stack. At the luminal surface, we observe some ZO-1 signal on the entire apical membrane, consistent with the images observed by Jafari and Rohn <https://doi.org/10.3389/fcimb.2023.1128132>. Deeper into the tissue, the ZO-1 staining is at cell junctions as expected. Data for different Z-slices at positions deeper into the tissue are shown in **Figure R1**. The ZO-1 signal (grayscale) is clearly evident in the cellular junctions only.

Figure R1 – Representative Z-slices within the tissue shown in Supplementary Fig 2H. ZO-1 is shown in grayscale.

- 4. Additional Comment:** In the new Supplementary Figure 2A and E, mature umbrella cells would typically appear as large polygonal cells positive for both UP3A and UP1A. In the new images, UP1A does not appear restricted to the superficial layer, and the expected large umbrella cells are not evident in panel D. Although some polygonal cells are visible in panel B, they do not seem large enough to represent fully mature umbrella cells. This is acceptable as no model is perfect, but it should be acknowledged as a limitation that the umbrella cell layer does not appear fully differentiated.

AU response: We have modified the sentence in the Discussion pertaining to the limitations of the model to address these concerns of the reviewer.

Line 425 - 'These include a variability in the number of cell layers and umbrella cell size, the absence of widespread AUM plaques and uroplakin staining uniformly lining the lumen, and the relatively higher cellular proliferation rate.'